# Digital quantum simulation of Floquet symmetry-protected topological phases

Xu Zhang[1,11], Wenjie Jiang[2,11], Jinfeng Deng[1,11], Ke Wang[1], Jiachen Chen[1], Pengfei Zhang[1], Wenhui Ren[1], Hang Dong[1], Shibo Xu[1], Yu Gao[1], Feitong Jin[1], Xuhao Zhu[1], Qiujiang Guo[1,3], Hekang Li[1,3], Chao Song[1,3], Alexey V. Gorshkov[4], Thomas Iadecola[5,6], Fangli Liu[4,7], Zhe-Xuan Gong[8,9], Zhen Wang[1,3 ✉], Dong-Ling Deng[2,10 ✉] & H. Wang[1,3]

Quantum many-body systems away from equilibrium host a rich variety of exotic phenomena that are forbidden by equilibrium thermodynamics. A prominent example is that of discrete time crystals[1–8], in which time-translational symmetry is spontaneously broken in periodically driven systems. Pioneering experiments have observed signatures of time crystalline phases with trapped ions[9,10], solid-state spin systems[11–15], ultracold atoms[16,17] and superconducting qubits[18–20]. Here we report the observation of a distinct type of non-equilibrium state of matter, Floquet symmetry-protected topological phases, which are implemented through digital quantum simulation with an array of programmable superconducting qubits. We observe robust long-lived temporal correlations and subharmonic temporal response for the edge spins over up to 40 driving cycles using a circuit of depth exceeding 240 and acting on 26 qubits. We demonstrate that the subharmonic response is independent of the initial state, and experimentally map out a phase boundary between the Floquet symmetry-protected topological and thermal phases. Our results establish a versatile digital simulation approach to exploring exotic non-equilibrium phases of matter with current noisy intermediate-scale quantum processors[21].

Symmetry-protected topological (SPT) phases are characterized by non-trivial edge states that are confined near the boundaries of the system and protected by global symmetries[22–26]. In a clean system without disorder, these edge states typically only occur for the ground states of systems with a bulk energy gap. At finite temperature, they are in general destroyed by mobile thermal excitations. However, adding strong disorder can make the system many-body localized (MBL)[27–31], allowing for a sharply defined topological phase and stable edge states even at infinite temperature[32–36]. Strikingly, the topological phase and corresponding edge states can even survive external periodic driving, as long as the driving frequency is large enough so that the localization persists[37,38].

The interplay between symmetry, topology, localization and periodic driving gives rise to various peculiar phases of matter that exist only out of equilibrium[38]. Understanding and categorizing these unconventional phases poses a well-known scientific challenge. On the theoretical side, topological classifications of periodically driven (Floquet) systems with[4,39–42] and without[43] interactions have already been obtained through a range of mathematical techniques (such as group cohomology), revealing a number of 'Floquet SPT' (FSPT) phases with no equilibrium counterparts[38]. Yet, we still lack powerful analytical tools or numerical algorithms to thoroughly address these phases and

their transitions to other ones. On the experimental side, signatures of discrete time crystals (DTCs)[1–8], which are paradigmatic examples of exotic phases beyond equilibrium[44], have been reported in a wide range of systems[9–20]. However, none of these experiments encompass topology as a key ingredient. A recent experiment simulating an FSPT phase on a trapped-ion quantum computer found that the phase was short-lived owing to the presence of coherent errors in the device[45]. Realizing a long-lived FSPT phase, which demands a delicate concurrence of topology, localization and periodic driving, thus still remains a notable experimental challenge.

Here we report the observation of non-equilibrium FSPT phases with a programmable array of 26 superconducting qubits (Fig. 1) with high controllability and long coherence time. We successfully implement the dynamics of prototypical time-(quasi)periodic Hamiltonians with $\mathbb{Z}_2 \times \mathbb{Z}_2, \mathbb{Z}_2$, or no microscopic symmetries, and observe subharmonic temporal responses for the edge spins. In particular, we focus on a one-dimensional (1D) time-periodic Hamiltonian with three-body interactions and $\mathbb{Z}_2 \times \mathbb{Z}_2$ symmetry as an example. We digitally simulate this Hamiltonian through a large-depth quantum circuit obtained using a neuroevolution algorithm[46]. We then measure local spin magnetizations and their temporal correlations and demonstrate that both quantities show a subharmonic response at the boundaries but not in the

[1]Department of Physics, ZJU-Hangzhou Global Scientific and Technological Innovation Center, Interdisciplinary Center for Quantum Information, and Zhejiang Province Key Laboratory of Quantum Technology and Device, Zhejiang University, Hangzhou, China. [2]Center for Quantum Information, IIIS, Tsinghua University, Beijing, China. [3]Alibaba-Zhejiang University Joint Research Institute of Frontier Technologies, Hangzhou, China. [4]Joint Quantum Institute and Joint Center for Quantum Information and Computer Science, University of Maryland and NIST, College Park, MD, USA. [5]Department of Physics and Astronomy, Iowa State University, Ames, IA, USA. [6]Ames Laboratory, Ames, IA, USA. [7]QuEra Computing Inc., Boston, MA, USA. [8]Department of Physics, Colorado School of Mines, Golden, CO, USA. [9]National Institute of Standards and Technology, Boulder, CO, USA. [10]Shanghai Qi Zhi Institute, Shanghai, China. [11]These authors contributed equally: Xu Zhang, Wenjie Jiang, Jinfeng Deng. ✉e-mail: 2010wangzhen@zju.edu.cn; dldeng@tsinghua.edu.cn

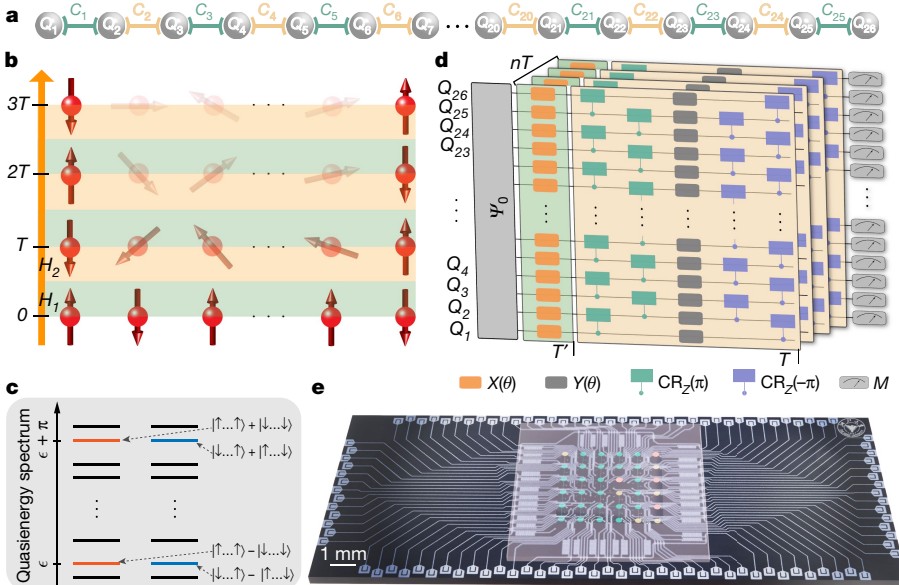

**Fig. 1 | FSPT phase and schematics of the experimental setup. a**, The 26 qubits used in our experiment are coupled to their neighbours with capacitive couplers. **b**, A chain of spins is periodically driven with the time-periodic Hamiltonian $H(t)$, giving rise to an FSPT phase characterized by time-translational symmetry breaking at the boundaries. **c**, The quasi-energy spectrum of the Floquet unitary $U_F$, which is the time evolution operator over one period. For the FSPT phase, every eigenstate with quasi-energy $\varepsilon$ is two-fold degenerate and has a cousin (denoted by the same colour) separated from it by quasi-energy $\pi$. Here, $|\uparrow\cdots\uparrow\rangle \pm |\downarrow\cdots\downarrow\rangle$ and $|\downarrow\cdots\uparrow\rangle \pm |\uparrow\cdots\downarrow\rangle$ denote the eigenstates of $U_F$ with $\delta = V_k = h_k = 0$

(Supplementary Information I.B). **d**, A schematic illustration of the experimental circuits used to implement the time dynamics governed by $H(t)$. We randomly sample the Hamiltonians and prepare the initial states as random product states or random static SPT states. After running a sequence of quantum gates, we measure the local magnetization or stabilizer operators at discrete time points. **e**, Illustration of the quantum processor, with the 26 qubits used in the experiment highlighted in green. Yellow circles are functional qubits, but not used owing to limited gate fidelity. The remaining lattice sites (denoted as red circles) are non-functional qubits.

bulk of the chain. This situation differs drastically from the case of DTCs, which exhibit subharmonic response everywhere in the bulk. This contrast stems from a fundamental distinction between DTC and FSPT phases: the former exhibit conventional long-range order in the bulk intertwined with the spontaneous breaking of discrete time-translational symmetry[44,47,48] whereas the latter exhibit SPT order that can only be revealed through boundary effects or non-local 'string operators' in the bulk[39,41,49]. The observed boundary subharmonic response persists over an extended range of parameters and is robust to various experimental imperfections, independent of the initial states. We further explore the FSPT phase experimentally from the perspectives of entanglement dynamics, the entanglement spectrum and the dynamics of stabilizer operators that underlies its topological nature. By measuring the variance of the subharmonic peak height in the Fourier spectrum, we experimentally map out the phase boundary between the FSPT and thermal phases.

## Model Hamiltonian and its implementation

We mainly consider a 1D spin-$\frac{1}{2}$ chain governed by the following time-periodic Hamiltonian (Fig. 1b):

$$H(t) = \begin{cases} H_1, & \text{for } 0 \leq t < T', \\ H_2, & \text{for } T' \leq t < T, \end{cases}$$

$$H_1 \equiv \left(\frac{\pi}{2} - \delta\right) \sum_k \hat{\sigma}_k^x, \tag{1}$$

$$H_2 \equiv -\sum_k [J_k \hat{\sigma}_{k-1}^z \hat{\sigma}_k^x \hat{\sigma}_{k+1}^z + V_k \hat{\sigma}_k^x \hat{\sigma}_{k+1}^x + h_k \hat{\sigma}_k^x],$$

where $\delta$ denotes the drive perturbation; $\hat{\sigma}_k^{x,z}$ is the Pauli matrix acting on the $k$th spin; $J_k$, $V_k$ and $h_k$ are random parameters drawn independently from uniform distributions over $[J - \Delta_J, J + \Delta_J]$, $[V - \Delta_V, V + \Delta_V]$

and $[h - \Delta_h, h + \Delta_h]$, respectively. For simplicity, we fix $T = 2T' = 2$, which roughly corresponds to 0.3 μs for running the corresponding quantum circuit in our experiment. We note that $H(t)$ has a $\mathbb{Z}_2 \times \mathbb{Z}_2$ symmetry. For a suitable parameter regime, it has been shown that $H_2$ can be in an MBL phase, in which topological edge states can survive as coherent degrees of freedom at arbitrarily high energies[34]. The localization and edge states carry over to the case of periodic driving with the Hamiltonian $H(t)$, giving rise to an FSPT phase. In this FSPT phase, the time-translational symmetry only breaks at the boundary but not in the bulk. The Floquet unitary that fully characterizes the FSPT phase reads $U_F = U_2 U_1$, where $U_1 = e^{-iH_1}$ and $U_2 = e^{-iH_2}$ are the unitary operators generated by the Hamiltonians $H_1$ and $H_2$, respectively. The quasi-energy spectrum of $U_F$ reveals that every eigenstate is two-fold degenerate and has a cousin eigenstate separated by the quasi-energy $\pi$ (Fig. 1c). The degenerate eigenstates also exhibit long-range mutual information between the boundary spins; this is essential for the robustness of the subharmonic response of the edge spins against local perturbations, including finite $\delta$ and $V_k$, that respect the $\mathbb{Z}_2 \times \mathbb{Z}_2$ symmetry (Methods and Supplementary Information I).

To implement $H(t)$ with superconducting qubits, the three-body term in $H_2$, which is crucial for the SPT phase at high energy, poses an apparent challenge because no three-body interaction appears naturally in the superconducting system. We thus use the idea of digital quantum simulation[50] to implement $H(t)$ with quantum circuits (Fig. 1d). For $V_k = h_k = 0$, we find optimal circuits in an analytical fashion that can implement $H(t)$ with arbitrary $J_k$ and $\delta$, whereas for non-vanishing $V_k$ and $h_k$ we use a neuroevolution algorithm[46] to design suitable quantum circuits (Methods). With the obtained quantum circuits, we perform our experiment on a flip-chip superconducting quantum processor (Fig. 1e) with a chain of $L = 26$ transmon qubits denoted as $Q_1$ to $Q_L$ (Fig. 1a). See Methods and Supplementary Information for the details of the experimental setup, and for experimental results from another processor with a chain of 14 qubits.

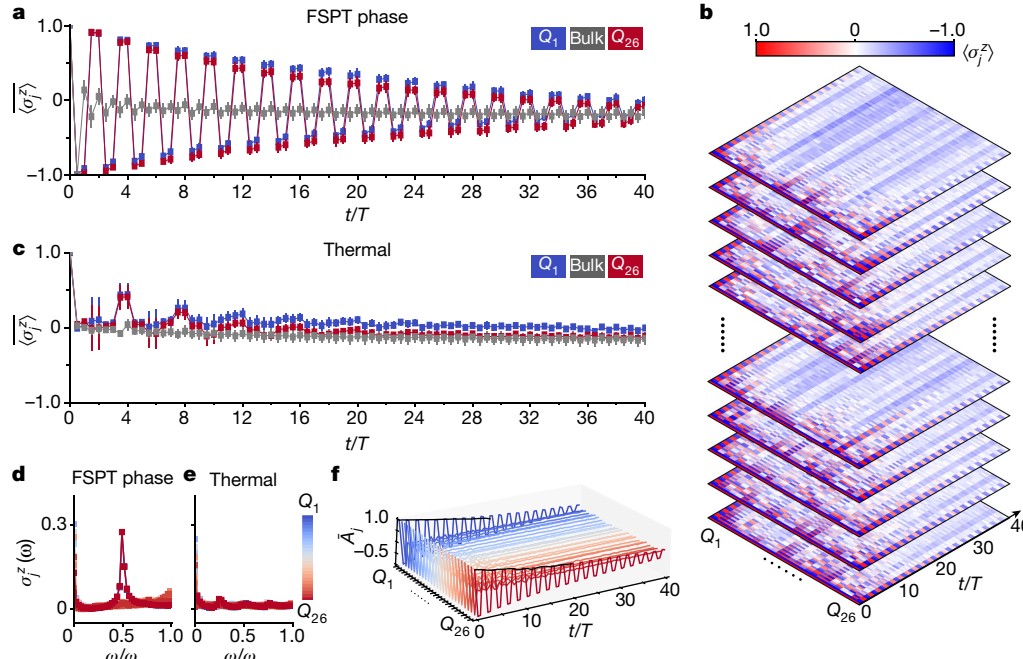

**Fig. 2 | Observation of an FSPT phase with 26 programmable superconducting qubits. a**, Time evolution of disorder-averaged local magnetizations deep in the FSPT phase ($L = 26$, $J = \Delta_J = 1$, $V = h = \Delta_V = \Delta_h = 0$ and $\delta = 0.01$). The initial state is $|0\rangle^{\otimes L}$, and the data shown are averaged over 20 random disorder instances. The error bars represent the standard error of the mean over disorder samples. Whereas the bulk magnetization decays quickly to zero, the edge spins oscillate with a stable subharmonic response for up to 40 cycles. **b**, The evolution dynamics of local magnetizations for different random instances. Here each layer corresponds to a specific random instance. **c**, Magnetization dynamics deep in the thermal phase ($J = \Delta_J = 1$, $V = h = \Delta_V = \Delta_h = 0$

and $\delta = 0.8$). **d**, Fourier transform of experimentally measured $\overline{\langle\sigma_j^z(t)\rangle}$ in the FSPT phase. The edge spins lock to the subharmonic frequency, which is in sharp contrast to the bulk spins. **e**, Fourier spectra of $\overline{\langle\sigma_j^z(t)\rangle}$ in the thermal phase. No robust subharmonic frequency peak appears for either edge spins or bulk spins in this case. **f**, Time dependence of the autocorrelator $\overline{A}_j = \overline{\langle\Psi_0|\sigma_j^z(t)\sigma_j^z(0)|\Psi_0\rangle}$ for up to 40 cycles, obtained from averaging over 20 random instances deep in the FSPT phase, with the initial states prepared as random product states in the computational basis. The black solid lines show the results of 'echo' circuits for the two boundary qubits.

## Symmetry breaking at boundaries

The characteristic signature of an FSPT phase is the breaking of the discrete time-translational symmetry at the boundaries of the chain but not in the bulk. This can be manifested by the persistent oscillation with period $2T$ of local magnetizations at the boundaries. In Fig. 2, we plot the time evolution of the disorder-averaged local magnetizations $\overline{\langle\sigma_j^z(t)\rangle}$ for different phases. From Fig. 2a, it is evident that in the FSPT phase, the disorder-averaged magnetizations at the two ends of the chain, namely $\overline{\langle\sigma_1^z(t)\rangle}$ and $\overline{\langle\sigma_L^z(t)\rangle}$, oscillate with a $2T$ periodicity, for up to 40 driving cycles. In stark contrast, the local magnetizations in the bulk of the chain ($\overline{\langle\sigma_j^z(t)\rangle}$ with $2 \leq j \leq L - 1$) decay quickly to zero and do not show period-doubled oscillations. This unconventional behaviour is independent of disorder averaging. Even for a single random disorder instance the magnetizations exhibit similar dynamical features, as shown in Fig. 2b. The distinction between the dynamics of boundary and bulk magnetizations can also be clearly seen by examining $\overline{\langle\sigma_j^z(t)\rangle}$ in the frequency domain. As shown in Fig. 2d, the edge spins lock to the subharmonic frequency of the drive period $\omega/\omega_0 = 1/2$, whereas the bulk spins show no such peak. We stress that the subharmonic response for the edge spins obtained in our experiment is notably robust to various perturbations (including non-zero $\delta$) and experimental imperfections (see Supplementary Information I.B for a more in-depth discussion). For comparison, we also experimentally measure the dynamics of the magnetizations in the thermal phase. Our results are shown in Fig. 2c,e, where we see that the magnetizations for both the edge and bulk spins decay quickly to zero and no subharmonic response appears at all.

The breaking of the discrete time-translational symmetry at the boundaries can also be detected by the disorder-averaged autocorrelators defined as $\overline{A}_j = \overline{\langle\Psi_0|\sigma_j^z(t)\sigma_j^z(0)|\Psi_0\rangle}$. Our experimental measurements of autocorrelators for up to 40 driving cycles are plotted in Fig. 2f, again showing the breaking of time-translational symmetry at the boundaries but not in the bulk. We mention that, in the FSPT phase, the local magnetizations for the edge spins exhibit a gradually decaying envelope, which could be attributed to either external circuit errors (that is, experimental imperfections such as decoherence, pulse distortions and cross-talk effects) or slow internal thermalization (namely, an intrinsic tendency towards thermalization in the model). To distinguish these two mechanisms, we carry out an additional experiment on the echo circuit $U_{echo} \equiv (U_F^\dagger)^t U_F^t$, the deviation of which from the identity operator measures the effect of circuit errors[18]. The square root of the output of $U_{echo}$ (black solid lines shown in Fig. 2f) fits well with the decaying envelope of the results obtained by evolution under $U_F$. This indicates that the decay of the envelope is due to circuit errors rather than thermalization, which corroborates that the system is indeed in the localized phase.

## Localization-protected topological states

In the above discussion, the initial states are random product states. To establish the FSPT phase, additional experiments on other initial states and other local observables are necessary. In this section, we show that the stabilizers in the bulk do not break the discrete time-translational symmetry, but at the boundaries they do. To understand this, we consider the idealized cluster-state and spin-flip limit, that is, $V_k = h_k = 0$ and $\delta = 0$.

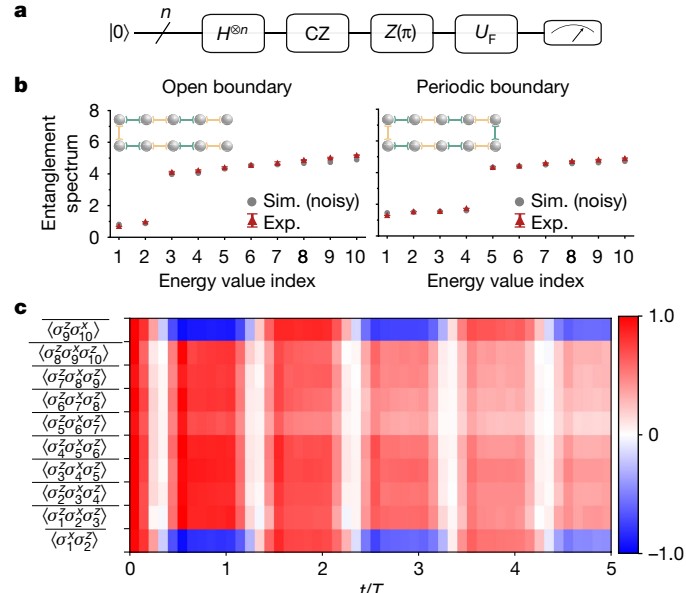

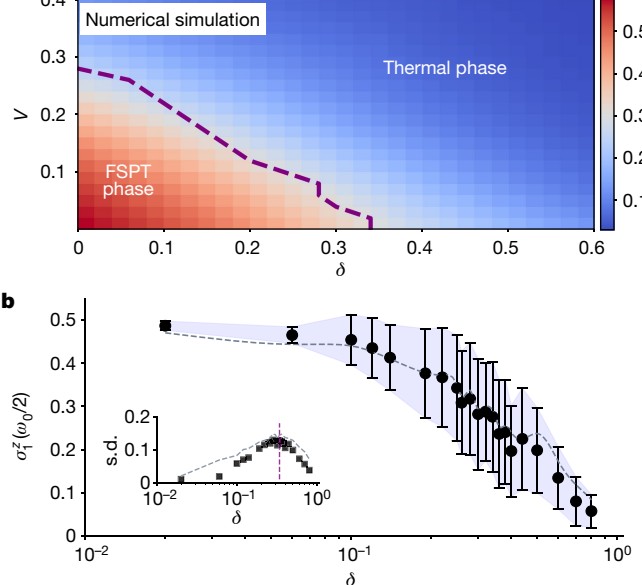

**Fig. 3 | Dynamics of stabilizers with random initial SPT states. a**, Schematic of the experimental circuit for preparing random SPT states. To prepare the system in the ground state of the stabilizer Hamiltonian $H_s$, we apply a Hadamard gate (H) on each qubit and then run CZ gates in parallel on all neighbouring qubit pairs in two steps. Then we apply $Z$ operators on random sites to create excitations, thus transferring the ground state to a highly excited eigenstate of $H_s$. This procedure enables the preparation of random SPT states at high energy. We then evolve these states with the Hamiltonian $H(t)$ to study the dynamics of stabilizers. **b**, Entanglement spectrum of a random SPT state evolved by one driving period, with open (left) and periodic (right) boundary conditions. The 'Energy value index' labels the eigenvalues of $-\ln(\rho_{\text{half}})$. The red triangles with error bars are the experimental (Exp.) results and the grey dots show the numerical simulations (Sim.) that take into account experimental imperfections (Supplementary Information IV). The two- and four-fold degeneracy (in the case of open and periodic boundary conditions, respectively) of the low-lying entanglement levels is a characteristic feature of the topological nature of these states. **c**, The time dependence of stabilizers in the FSPT phase, averaged over 20 random circuit instances. The parameters in **b** and **c** are chosen as $L = 10$, $\delta = 0.1$, $J = \Delta_j = 1$, $h = \Delta_h = 0.01$ and $V = \Delta_v = 0.01$.

**Fig. 4 | Numerical phase diagram and experimental detection of the phase transition. a**, The numerical $\delta - V$ phase diagram obtained by examining the central subharmonic peak height for the edge spins in the Fourier spectrum, averaged over 50,000 disorder instances. The dashed line corresponds to the maximal height variances for varying $V$ with each fixed $\delta$ point, which gives a rough estimation of the phase boundary. Here, the parameters are chosen as $L = 8$, $J = \Delta_j = 1$ and $h = \Delta_h = \Delta_v = 0$. **b**, Experimental result of the subharmonic peak height as a function of $\delta$ with fixed $h = V = \Delta_h = \Delta_v = 0$ and $J = \Delta_j = 1$, averaged over 50 disorder instances uniformly sampled from the interval $[J - \Delta_j, J + \Delta_j]$ for $L = 8$ qubits, with the shadow outlining the standard deviation. Inset: the standard deviation of the central peak height as a function of $\delta$. The purple vertical dashed line at $\delta = 0.34$ indicates the numerically determined phase boundary from **a**. The dashed lines in grey show numerical simulations with experimental noises taken into account.

In this limit, $H_2$ reduces to a summation of stabilizers: $H_s = -\sum_{k=2}^{L-1} J_k S_k$ with $S_k \equiv \hat{\sigma}_{k-1}^z \hat{\sigma}_k^x \hat{\sigma}_{k+1}^z$. We choose the initial states to be random eigenstates of $H_s$ and evolve the system with the time-periodic Hamiltonian $H(t)$ to measure the time dependence of local stabilizers.

In Fig. 3a, we show a sketch of the quantum circuit used in our experiment to prepare the desired random eigenstates of $H_s$. To manifest the topological nature of these eigenstates, we study their entanglement spectra[51], which are widely used as a crucial diagnostic for universal topological properties of quantum phases[51–54]. To show that $H(t)$ preserves the topological nature of the SPT states, we prepare random eigenstates of $H_s$ with both open and periodic boundary conditions, evolve the system for one driving period with $H(t)$ and then measure the reduced density matrix $\rho_{\text{half}}$ of half of the system through quantum-state tomography. Figure 3b displays the entanglement spectra (eigenvalues of $-\ln(\rho_{\text{half}})$) for open and periodic boundary conditions, respectively. From this figure, a clear two-fold degeneracy for the low-lying Schmidt states is obtained for the open boundary conditions. This degeneracy corresponds to an effectively decoupled spin-half degree of freedom at the boundary of the bipartition. For periodic boundary conditions, the spectrum is four-fold degenerate, corresponding to two effectively decoupled spins at the two boundaries of the bipartition. The degeneracy of the entanglement spectrum and its dependence on boundary conditions marks a characteristic

feature of the SPT state generated in our experiment. We note that the degeneracy disappears above the entanglement gap. This is due to finite-size effects and experimental imperfections.

In Fig. 3c, we plot the time dependence of local stabilizers in the FSPT phase. We observe that the stabilizers at the boundaries oscillate with a $2T$ periodicity, indicating again the breaking of discrete time-translational symmetry at the boundaries. In the bulk, the stabilizers oscillate with a $T$ periodicity and are synchronized with the driving frequency, showing that no symmetry breaking occurs. This is in sharp contrast to the dynamics of bulk magnetizations, which decay rapidly to zero and exhibit no oscillation, as shown in Fig. 2a. In fact, in the FSPT phase, the system is MBL and there exist a set of local integrals of motion, which are the 'dressed' versions of the stabilizers with exponentially small tails[34]. The persistent oscillations of the bulk stabilizers observed in our experiment originate from these local integrals of motion and are a reflection of the fact that the system is indeed in an MBL phase.

## Phase transition

We now turn to the phase transition between the FSPT phase and the trivial thermal phase. For simplicity and concreteness, we fix other parameters and vary the drive perturbation $\delta$ and the interaction strength $V$. Theoretically, the system is expected to exhibit an FSPT phase for small $\delta$ and $V$. With increasing $\delta$ and $V$, the strong interaction diminishes localization and eventually thermalizes the system. At some critical values of $\delta$ and $V$, a transition between these two phases occurs. In Fig. 4a, we plot the $\delta - V$ phase diagram obtained from numerical

simulations, in which the phase boundary, although not very sharp because of finite-size effects (for a small system size $L = 8$, the coupling between the two edge modes is not negligible and thus will decrease the central subharmonic peak height and result in a blurred boundary), can be located and visualized approximately.

To experimentally examine this phase transition, we further fix the interaction strength $V = 0$. We probe the transition point by measuring the variance of the subharmonic spectral peak height, that is, the amplitude of the Fourier spectrum of $\overline{\langle \sigma_1^z(t) \rangle}$ at $\omega = \omega_0/2$ for the boundary spin. Figure 4b shows the subharmonic peak height as a function of the drive perturbation $\delta$. At small $\delta$, the system is in the FSPT phase, and the peak height remains at a value around 0.5. As we increase $\delta$ to a large value, the system transitions out of the topological phase and the peak height vanishes. This is consistent with the theoretical analysis above. The largest variance of the peak height corresponds to the phase transition point. The inset of Fig. 4b shows the measured standard deviation as a function of $\delta$, indicating a phase transition point around $\delta \approx 0.30$, which is consistent with the numerically predicted value of 0.34. The small deviation between the numerical prediction and experimental result is mainly attributed to finite-size effects, experimental noise and the limited number of disorder instances implemented in the experiment.

## Other non-equilibrium SPT phases

The digital simulation approach used in our experiment is generally applicable for quantum simulations of various exotic phases of matter. The model Hamiltonian in equation (1) possesses a $\mathbb{Z}_2 \times \mathbb{Z}_2$ symmetry, which can also support robust edge modes in the static equilibrium setting. For driven non-equilibrium systems, however, the edge modes may be stabilized by emergent dynamical symmetries. To demonstrate this and illustrate the general applicability of our approach, we also digitally simulate two other models with our quantum device, namely a periodically driven Ising chain with $\mathbb{Z}_2$ symmetry and a quasiperiodically driven model without any microscopic symmetry (Methods and Supplementary Information VI). Our results are summarized in Extended Data Figs. 1 and 2, in which robust subharmonic edge oscillations are also observed.

## Conclusions

In summary, we have experimentally observed signatures of non-equilibrium Floquet SPT phases with a programmable superconducting quantum processor. In contrast to previously reported conventional time crystals, for our observed FSPT phases, the discrete time-translational symmetry only breaks at the boundaries and not in the bulk. We measured the persistent oscillations of edge spins with a subharmonic frequency and experimentally demonstrated that the FSPT phases are robust to symmetry-respecting perturbations in the drive and imperfections in the experiment. In addition, we also demonstrated that the subharmonic response of boundary observables is independent of the initial state. The digital quantum simulation approach explored in our experiment is generally applicable to the simulation of a wide range of non-equilibrium systems hosting unconventional topological phases, including those with multi-body interactions.

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

# Article

## Methods

### Characterization of the model Hamiltonian

To understand why time-translational symmetry breaks at the boundary but not in the bulk, we consider the idealized 'cluster-model' limit ($V_k = h_k = 0$) and set $\delta = 0$. We suppose that the system is initially prepared in a random product state in the computational basis, and we use the dynamics of local magnetization as a diagnostic. In this simple scenario, the topologically non-trivial structure of the cluster states (eigenstates of $U_2$) gives rise to edge modes that behave as free spins. At each driving period, the unitary operator $U_1$ flips all spins. As a result, the edge spins are reversed after one period and return to their initial configuration after two, leading to the period-doubled dynamics of the local magnetization at the boundaries. For spins in the bulk, however, the unitary operator $U_2$ plays a part and evolves the random product state to a state with vanishing magnetization, resulting in no period doubling. When $V_k = 0$, the Hamiltonian in equation (1) can be mapped to free Majorana fermions (Supplementary Information I.B and, for example, refs. [55,56]). Further setting $\delta = h_k = 0$, we find that equation (1) maps onto two decoupled copies of the fixed-point model of a $\mathbb{Z}_2$ FSPT phase considered in ref. [39]. The robustness of the subharmonic responses of the topologically protected edge spins to perturbations respecting the $\mathbb{Z}_2 \times \mathbb{Z}_2$ symmetry is discussed in depth in Supplementary Information I.B.

### Logarithmic entanglement growth

For a MBL system, the entanglement entropy will feature a logarithmic growth[57], which is in sharp contrast to the case of Anderson localization without interactions. For the model Hamiltonian $H(t)$ studied in this Article, we also expect a logarithmic growth of the entanglement entropy inside the FSPT phase with $V_k \neq 0$. We numerically simulate the entanglement dynamics of the system deep in the FSPT phase with the time-evolving block decimation algorithm up to a system size $L = 100$ (Supplementary Information II). Our results clearly verify the logarithmic entanglement growth, which again implies that the FSPT phase is indeed MBL with non-vanishing $V_k$. In our experiment, we also study the entanglement dynamics for a small system size ($L = 6$) through quantum tomography (Supplementary Information V). We find that in the thermal phase the entanglement grows much faster than that in the FSPT phase. However, because of the small system size and experimental imperfections (such as decoherence, pulse distortions and cross-talk effects), we are not able to observe the logarithmic entanglement growth (Supplementary Information VI).

### Quantum circuits for implementing $H(t)$

Direct implementation of the Floquet Hamiltonian $H(t)$ with superconducting qubits faces a notable difficulty: the natural interactions hosted by the superconducting qubits are only two-body, so the three-body terms in $H_2$ cannot emerge directly. Fortunately, programmable superconducting qubits are universal for quantum computation; thus we can explore the idea of digital quantum simulation to emulate the dynamics of $H(t)$. However, because of inevitable experimental imperfections, the depth of the quantum circuits is limited. As a result, obtaining well-performing circuits with an optimal depth that can implement $H(t)$ (or equivalently the Floquet unitary $U_f$) is of crucial importance for the success of our experiment.

To find the desired quantum circuits, we use a neuroevolution method introduced in ref. [46], which outputs a near-optimal architecture for a family of variational quantum circuits that can implement $H(t)$ with different random disorder instances. For a given instance of $J_k$, $V_k$ and $h_k$, we use the gradient decent method to tune the variational parameters of the ansatz circuits to minimize the distance between the unitary represented by the circuit and the unitary generated by $H(t)$ within a small time interval. In the idealized 'cluster-model' limit ($V_k = h_k = 0$), we can find a simple exact one-to-one correspondence between $J_k$ and the variational parameters, independent of the system size and the values of $J_k$ and $\delta$. Thus, we are able to construct an analytical quantum circuit (see Supplementary Fig. 4c for an explicit illustration of the circuit for $L = 6$) that can implement $H(t)$ precisely and, at the same time, in a way that is experimentally friendly and practical. The details of how to obtain the desired quantum circuits are given in Supplementary Information III.

### Experimental setup

Our experiment is performed on a flip-chip superconducting quantum processor designed to encapsulate a square array of $6 \times 6$ transmon qubits with adjustable nearest-neighbour couplings (Fig. 1e), on which a chain of up to $L = 26$ qubits, denoted as $Q_1$ to $Q_L$, that alternate with $L - 1$ couplers, denoted as $C_1$ to $C_{L-1}$, are selected to observe the FSPT phase (Fig. 1a). All $L$ qubits can be individually tuned in frequency with flux biases, excited by microwaves, and measured using on-chip readout resonators; all couplers are also of transmon type with characteristic transition frequencies higher than those of the qubits, which can be controlled with flux biases to tune the effective nearest-neighbour couplings. During an experimental sequence (Fig. 1d), we first initialize each qubit, $Q_j$, in $|0\rangle$ at its idle frequency $\omega_j$, following which we alternate the single-qubit gates at $\omega_j$ with the two-qubit controlled-$\pi$ (CZ) gates realized by biasing $Q_j$ and its neighbouring qubit to the pairwise frequencies of group A(B) listed in ($\omega_j^{A(B)}$, $\omega_{j+1}^{A(B)}$) for a fixed interaction time (Supplementary Information III.C). Meanwhile, each coupler is dynamically switched between two frequencies[58–63]: one is to turn off the effective coupling where the neighbouring two qubits can be initialized and operated with single-qubit gates; the other one is to turn on the nearest-neighbour coupling to around 11 MHz for a CZ gate. After $n$ layers of the alternating single- and two-qubit gates, we finally tune all qubits to their respective $\omega_j^m$ (here, the superscript 'm' stands for 'measurement') for simultaneous quantum-state measurement. Qubit energy relaxation times measured around $\omega_j$ are in the range of 7–41 μs, averaging above 30 μs. More characteristic qubit parameters, including the above mentioned frequencies, anharmonicities and readout fidelities, can be found in Supplementary Table 1. The parameters for another processor with 14 qubits used are displayed in the Supplementary Table 2.

We explore a quantum digital simulation scheme to implement the dynamics of the system under the driven Hamiltonian $H(t)$. More specifically, we decompose the evolution operators into the experimentally feasible single-qubit gates ($X(\theta)$, $Y(\theta)$ and $Z(\theta)$) and two-qubit gates ($CR_z(\pm\pi)$), where $X(\theta)$, $Y(\theta)$ and $Z(\theta)$ are rotations around the $x$, $y$ and $z$ axes by the angle $\theta$, respectively, and $CR_z(\pm\pi)$ are the $z$-axis rotations of the target qubit by $\pm\pi$ conditioned on the state of the control qubit (Fig. 1d and Supplementary Information III.A for the ansatz that generates the gate sequences). Here $X(\theta)$ and $Y(\theta)$ are realized by applying 50-ns-long microwave pulses with a full-width half-maximum of 25 ns, for which the quadrature correction terms are optimized to minimize state leakages to higher levels[64]. Simultaneous randomized benchmarkings indicate that the single-qubit gates used in this experiment have reasonably high fidelities, averaging above 0.99 (Supplementary Table 1). Then $Z(\theta)$ is realized using the virtual-Z gate, which encodes the information $\theta$ in the rotation axes of all subsequent gates[65], and is combined with CZ to assemble $CR_z(\pm\pi)$. Here we adopt the strategy reported elsewhere[62,66] to realize the CZ gate, that is, we diabatically tune the coupler frequency while keeping $|11\rangle$ and $|02\rangle$ (or $|20\rangle$) for the subspace of the two neighbouring qubits in near resonance. When simultaneously running the 40-ns-long CZ gates for multiple pairs of neighbouring qubits as required in the experimental sequence, the average CZ gate fidelities can be above 0.98, as obtained by simultaneous randomized benchmarking (Supplementary Table 1).

### Further experiments on non-equilibrium SPT phases

The digital simulation strategies of our experiments are capable of simulating a wide range of models hosting unconventional non-equilibrium topological phases. To illustrate this, we also implement two other dynamical SPT phases with our superconducting quantum processor:

an FSPT phase in a periodically driven random Ising chain[4] and an emergent dynamical SPT (EDSPT) phase in a quasiperiodically driven chain[67].

The first model has a $\mathbb{Z}_2$ (Ising) symmetry. For the FSPT phase (ref. [4] and Supplementary Information VI.A), the evolution is realized by applying two unitaries in an alternating fashion (Extended Data Fig. 1a) to random initial states. For the parameters chosen in our experiments, the corresponding Floquet unitary $U_F = e^{-iH_{Ising}}e^{-iH_{single}}$, where $H_{single} = \sum_k g_k \hat{\sigma}_k^x$ and $H_{Ising} = \sum_k J_k \hat{\sigma}_k^z \hat{\sigma}_{k+1}^z$ with $g_k$ and $J_k$ being coupling parameters respectively, maintains a $\mathbb{Z}_2 \times \mathbb{Z}$ symmetry (where $\mathbb{Z}$ describes discrete time-translation symmetry), despite the fact that the original static Hamiltonian only possesses a $\mathbb{Z}_2$ symmetry. This enlarged dynamical symmetry protects the edge modes of this phase, one at quasi-energy 0 and the other at quasi-energy π. This leads to unusual dynamics of the edge spins. If one applies this evolution to a product state in the $x$ basis, the edge spins will return to their initial states only at even periods. In our experiments, we measure the random disorder-averaged local magnetization $\overline{\langle \sigma_k^x \rangle}$ during the evolution (Extended Data Fig. 1b). Persistent subharmonic oscillations are observed for the edge spins, whereas the averaged magnetizaiton in the bulk is synchronized with the driving frequency and shows no breaking of the discrete time-translational symmetry.

The EDSPT model has no microscopic symmetry (see refs. [45,67] and Supplementary Information VI.B). The evolution of an initial state is realized by applying on it a sequence of evolution unitaries at Fibonacci times, $U^{(v)} = U(t_v = F_v)$, with $F_v$ being the $v$th element of the Fibonacci sequence. Although the underlying Hamiltonian of this model includes random fields breaking all microscopic symmetries, the evolution unitary possesses a locally dressed $\mathbb{Z}_2 \times \mathbb{Z}_2$ symmetry emergent from the quasiperiodic drive[45,67]. The emergent symmetry hosts two non-trivial edge modes, which can be manifested by the distinct dynamics of the edge spins. In particular, the edge spins would exhibit $3v$-periodic oscillations when measured at Fibonacci times $t_v = F_v$, whereas the magnetization of the bulk spins will decay to zero rapidly. In our experiment, we prepare random initial states and use the circuits shown in Extended Data Fig. 1a,b to implement the quasiperiodic driving of the system. We measure the random disorder-averaged magnetizations $\overline{\langle \sigma_j^z \rangle}$ and $\overline{\langle \sigma_j^x \rangle}$ at Fibonacci times. Our experimental results are summarized in Extended Data Fig. 2c, in which persistent quasiperiodic oscillations for edge spins are indeed observed. The results shown in Extended Data Figs. 1 and 2 were obtained using 12 qubits on a third device with slightly improved performance. We note that an experimental implementation of the EDSPT model with ten trapped-ion qubits has recently also been reported[45].

## Data availability

The data presented in the figures and that support the other findings of this study are available for download at https://doi.org/10.5281/zenodo.6510867.

## Code availability

The data analysis and numerical simulation codes are available at https://doi.org/10.5281/zenodo.6510867.

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

**Acknowledgements** We thank L.M. Duan and S.L. Xu for helpful discussion. The device was fabricated at the Micro-Nano Fabrication Centre of Zhejiang University. X. Zhang, J.D., K.W., J.C., P.Z., W.R., H.D., S.X., Y.G., F.J., X. Zhu, Q.G., H.L., C.S., Z.W. and H.W. acknowledge the support of the National Natural Science Foundation of China (grant nos. 11725419, U20A2076 and 92065204), the National Basic Research Programme of China (grant no. 2017YFA0304300), the Zhejiang Province Key Research and Development Programme (grant no. 2020C01019) and the Key-Area Research and Development Programme of Guangdong Province (grant no. 2020B0303030001). W.J. and D.-L.D. are supported by the National Natural Science Foundation of China (grant no. 12075128), Tsinghua University and the Shanghai Qi Zhi Institute. A.V.G. and F.L. acknowledge funding by the AFOSR, DoE QSA and NSF QLCI (award no. OMA-2120757).

**Author contributions** X. Zhang and J.D. carried out the experiments under the supervision of Z.W. and H.W. H.L., Z.W. and J.C. fabricated the device supervised by H.W. K.W., P.Z., W.R., H.D., S.X, Y.G., F.J., X. Zhu, Q.G. and C.S. characterized and calibrated the device. W.J. performed the numerical simulations under the supervision of D.-L.D. A.V.G., T.I., F.L., Z.-X.G. and D.-L.D. conducted the theoretical analysis. All authors contributed to the experimental setup, the discussions of the results and the writing of the manuscript.

**Competing interests** The authors declare no competing interests.

**Additional information**
**Correspondence and requests for materials** should be addressed to Zhen Wang or Dong-Ling Deng.

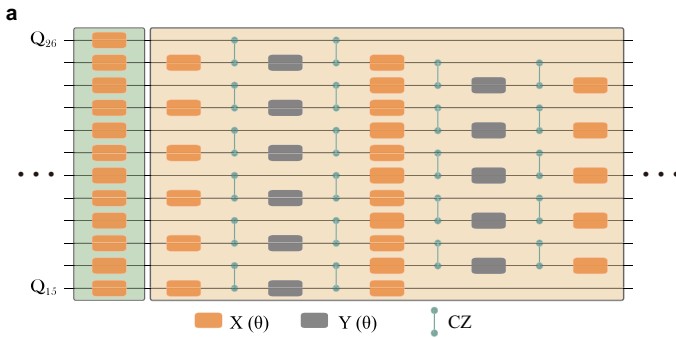

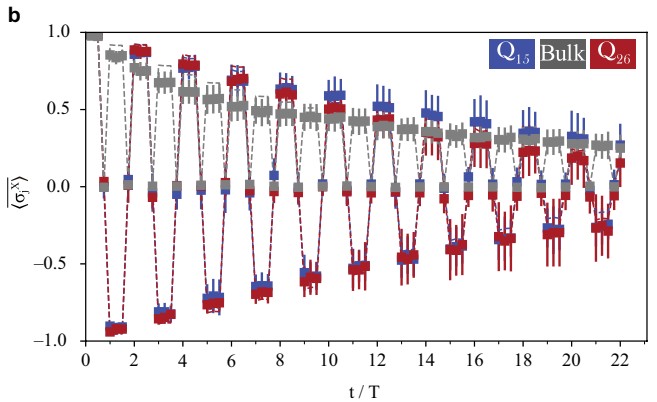

**Extended Data Fig. 1 | FSPT phase protected by a $\mathbb{Z}_2$ symmetry. a** Schematic of the experimental circuits for the Floquet unitary, where the first layer represents the evolution under the one-body Hamiltonian $H_{single}$, and the following layers represents the evolution under the Ising Hamiltonian $H_{Ising}$. **b** The dynamics of the edge and bulk magnetizations. Here, the expectation values of $\hat{\sigma}_j^x$ are multiplied by the signs of those of the random initial product states prepared in the $\hat{\sigma}^x$ basis. The edge magnetizations are averaged over 12 random disorder realizations, and the bulk magnetizations are averaged over 12 random realizations and all bulk sites. The dashed lines show numerical simulations with experimental noises. Here, we set $J_k = \frac{\pi}{2} - 0.1$, and choose $g_k$ uniformly from $\left[\frac{\pi}{4} - \frac{\pi}{3}, \frac{\pi}{4} + \frac{\pi}{3}\right]$ (Supplementary Information VI.A).

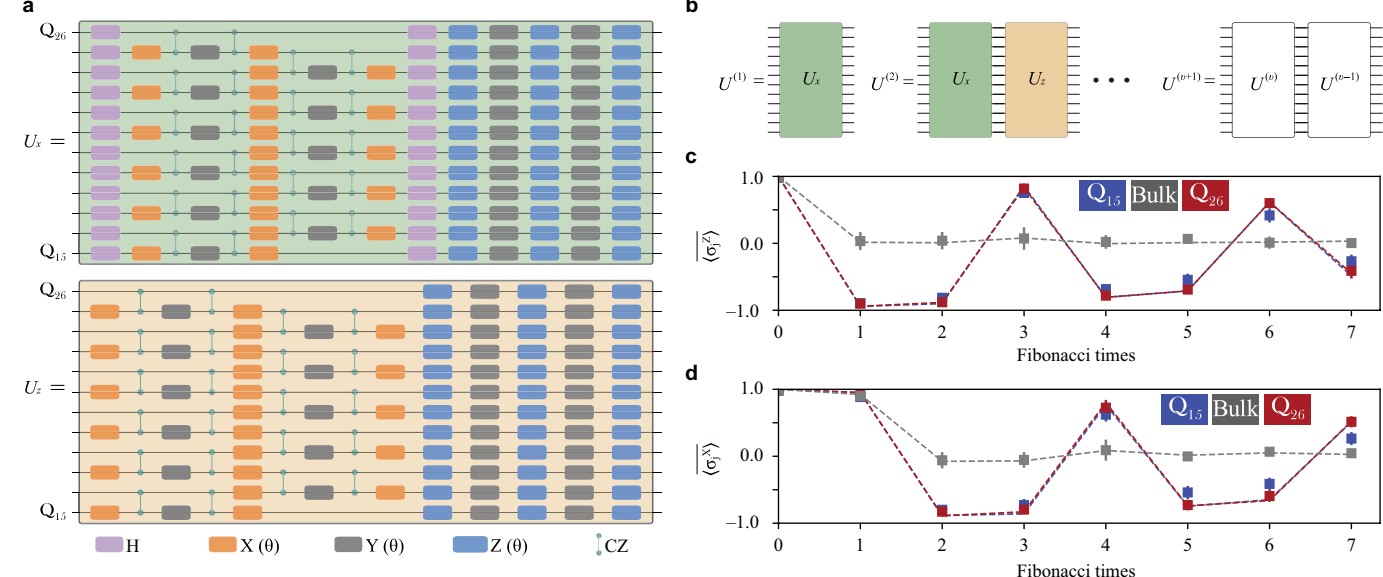

**Extended Data Fig. 2 | EDSPT phase protected by emergent dynamical symmetry. a** Schematic of the experimental circuits for implementing $U_x$ and $U_z$ respectively, which are the building blocks of the quasiperiodically-driven EDSPT model. **b** The circuit implementations of the evolution unitary $U^{(v+1)} = U^{(v-1)}U^{(v)}$ that defines the EDSPT model. **c**, **d** The dynamics of the edge and bulk magnetizations. Here, the expectation values of $\hat{\sigma}_j^z$, $\hat{\sigma}_j^x$ are multiplied by the signs of those of the random initial states respectively. The edge magnetizations are averaged over 12 random disorder realizations and 10 random initial states, and the bulk magnetizations are averaged over 12 random disorder realizations, 10 random initial states, and all bulk sites. The dashed lines show numerical simulations taking into account experimental noise. The imperfect pulse is set as $J = 0.99\pi$. The coupling parameters $K_k^x$, $K_k^z$ are uniformly chosen from $[0, 4\pi]$. The norms of the fields $\mathbf{B}_k^x$, $\mathbf{B}_k^z$ are uniformly chosen from $[0, 0.3]$, and their random directions are also chosen uniformly (Supplementary Information VI.B).