## [Peer Review File · Nature]

Manuscript Title: Digital quantum simulation of Floquet symmetry-protected topological phases

Reviewer Comments & Author Rebuttals

Reviewer Reports on the Initial Version:

Referees' comments:

Referee #1 (Remarks to the Author):

Over the last couple of years, Floquet Time Crystals, periodically driven systems that break discrete time translation invariance, have been realized in various quantum computing and quantum simulation platforms. Those works have time-crystalline order both in the bulk and at the edge. Crucially, such dynamical phases of matter cannot be realized in equilibrium but only dynamically. In this work, a related concept of a Floquet Symmetry Protected Topological (FSPT) phase has been investigated with a superconducting qubit quantum computer. The essential properties are that time-translational symmetry is broken at the edge of the system, manifesting in periodic modulation of local observables at the edge, while in the bulk such order decays. An essential ingredient to stabilize these phases is many-body localization, which stabilizes the 'order'. In this work circuits are developed that demonstrate a FSPT phase and investigate the edge coherence.

Over the past couple of years, many theoretical works have studied the concept of FSPT phases and their generalization. This work is essentially an application of these ideas on a quantum computer. As such the novelty of the work is not directly evident. In particular, as the quantum computer works only with 14 qubits, all quantum computations (even including noise models) can be immediately simulated on classical resources. From a theoretical standpoint, therefore these quantum computations are nice to have but they don't help to further advance the field of non-equilibrium quantum phases.

The data analysis needs to be improved. Several figure captions don't describe what kind of error the error bars represent (vertical lines are assumed to represent error bars in Fig. 2). Most likely it is just the s.e.m. over disorder samples. Anyway, instead of me speculating, the authors should clarify what is shown. Moreover, the authors do not perform direct comparisons with exact numerical results. I would find such comparisons helpful in all figures. In particular in Fig. 4b it would be the most natural to have them as the numerical data is even shown in Fig. 4a. Also, the caption of Fig. 4 is therefore misleading. It should rather say "Numerical phase diagram ..." to avoid confusion.

One has to stare for quite a while on Fig. 1c to understand what the authors intent to show. This could be improved.

For Floquet protocols, one typically expects measurements after every completed period. It is not done in that way in Fig. 2a. This should be clearly discussed in the text to clarify the situation (perhaps the authors want to report and show only full periods?). Also x-axis ticks and labels on panel

2a would be useful.

Fig. 3b shows the entanglement spectrum of a reduced density matrix directly obtained after state preparation of a cluster state. While some of the eigenvalues are somewhat nearby, degeneracies are absent and the spectrum is rather different from what is expected theoretically. Given the low depth of the circuits it should be critically discussed where this discrepancy comes from (currently only a vague sentence is included in the main text). This could help to understand the limitations of their system. It would be also important to state how the authors extract the density matrix from the measurements. The subsystem is so small that it should be possible to learn something about the device from these imperfections.

Logarithmic entanglement growth is claimed to be present in the exact numerical results. Looking at Fig. S1c, I wonder where authors see logarithmic growth?

The conclusions of the work discuss the realization of the non-equilibrium phase with nontrivial topological properties. In the last paragraph, the authors state that 'The controllability and scalability of the superconducting platform demonstrated in our experiment...'. I don't think that this work is a strong showcase of scalability of superconducting qubits. I suggest to clarify this statement or remove it. Currently it does not carry any weight.

The theoretical concept of dynamical phases without any equilibrium analogue and with nontrivial topological properties is very interesting. These concepts have been introduced theoretically in previous works. I am not sure what this study can add to the current understanding. Somewhat open question in the field could be late time/large system effects, and certainly behavior in two dimensions etc. The current work is a nice demonstration on a quantum computer, but all results could be easily simulated on a classical computer. Therefore, this work does not seem to fulfill the stringent requirements of Nature but seems to be rather suited for a more specialized journal.

Referee #2 (Remarks to the Author):

The manuscript entitled "Observation of a Floquet symmetry-protected topological phase with superconducting qubits" by Zhang et al. presents a compelling demonstration of Floquet symmetry-protected topological (FSPT) phase of matter implemented digitally using a 1D array of superconducting qubits.

The main results of the manuscript are, first, the observation of the FSPT phase and the 2T periodicity of the edge qubits (1, 14) with essentially no net magnetization of the bulk qubits (2-13). This was observed for randomized input states. Second, the authors tested the time-dependence of local stabilizers, again showing 2T-periodicity at the edges, with T-periodicity in the bulk. Lastly, the authors demonstrate a transition from the FSPT phase to a thermal phase, realized by increasing the Hamiltonian parameters that reduced many-body localization. As I describe below, I have a few

suggestions on this part.

Overall, I found the manuscript straightforward to read and follow. The experiments are carefully documented and support the statements in the paper. I think the topic will be of interest to physicists and quantum engineers alike. Whether this belongs in Nature or Nature Physics is an editorial decision.

I have a few questions / suggestions for the authors to consider

1) In Fig. 1, there are green qubits and yellow qubits. The green qubits are the 14 used in this experiment. The yellow ones are unused qubits. The authors should mention the meaning of the yellow color scheme in the caption (they already mention the green). Also, not all lattice sites have yellow dots. Does this mean those qubits did not work?

2) The authors set $T = 2T_1 = 2$. I may have missed it, but it would be good to specify how long this is in ns or us. Also, perhaps use something to distinguish this T_1 from relaxation T_1 , e.g., $t_{1,1}$.

3) The authors talk about disorder averaging at the beginning of the Symmetry breaking section, but only explain that it means randomizing the input states at the beginning of the next section (Localization-protected topological states). I would state earlier what it means.

4) What is meant by “external circuit errors” and what is meant by “thermalization”? I would explain this terminology in a sentence or so.

5) In the Localization-protected topological states section, the authors say that the periodic oscillations of the edge (Fig3c) are in “sharp contrast to the dynamics of bulk magnetization, which decay rapidly to zero”. I did not see this so obviously in Fig. 3c. It may be worthwhile to take line cuts to make it clear. If I understood correctly, the dark red at short times becomes whiter toward $t/T = 5$, but it still seems quite red to me (at least on my computer). Maybe change the wording, or show how it is decaying, alter the color scale, etc.

6) I found the phase transition to be the least supported part of the manuscript. I have a few suggestions here. First off, make it clear that 4a is a numerical simulation in the caption. It is called numerical simulation in the main text. Then, I would suggest explaining why the phase boundary is not so sharp. You indicate it is because of finite size effects. Is there anything more to say about it to help the reader understand? Lastly, if I understand correctly, Fig 4b corresponds to $V=0$ in Fig 4a. If that is indeed the case, could you put a line cut from 4a in to 4b, so we can see how well it matches?

7) In the Methods, you stated that you could not observe the logarithmic entanglement growth. This is due to “decoherence and other experimental imperfections”. What are the experimental imperfections? I would be clear here. Do you mean gate fidelity is not high enough? Do you mean Trotterization and the available circuit depth don’t allow for it? Do you mean that characterizing entanglement via tomography scales very poorly with system size? All of the above? Do simulations using your decoherence and your experimental imperfections (what are they?) indicate that you do not expect to see the entanglement growth? The SM does not really say much more about this,

other than we need more efficient ways to measure the entanglement growth. I would expand the discussion a bit, certainly beyond “decoherence and other experimental imperfections”.

8) I would suggest the authors not use overused idioms like “paves the way” or “opens the door”. Can you think of more precise terms for what your work actually enables?

Referee #3 (Remarks to the Author):

The manuscript by Zhang et al. describes the realization of a Floquet symmetry protected topological phase using a one dimensional chain ($L=14$) of superconducting transmon qubits. The manuscript builds upon the recent excitement surrounding the discovery that there exist intrinsically non-equilibrium phases of matter that do not have an equilibrium counterpart. The experiment’s ability to faithfully realize a complex three-body ZXZ interaction is quite impressive, although very few details are given regarding how this is done. Of course, as the authors’ direct comparison to numerics already indicates, a 14 qubit quantum simulation can readily be done on a classical computer, so this itself is perhaps not the defining feature; on the other hand, I do think that the experimental demonstration of the underlying conceptual framework surrounding Floquet SPT phases (as well as the realization of complex many-body interactions) is something that has the potential to have wide-ranging impacts. However, to be totally honest, despite multiple readings of the manuscript, I remain confused about a number of key points and would very much appreciate the authors’ explanations.

Perhaps most importantly, the authors emphasize that they are reporting the observation of “an intrinsically non-equilibrium FSPT phase.” I am a bit confused whether this is actually correct. This question may get a bit long, but I want to be absolutely sure that I am being as clear as possible. In particular, the physics that the author’s are exploring can be understood by analogy to the equilibrium AKLT/Haldane spin chain, which exhibits a decoupled spin-1/2 edge degree of freedom. In the equilibrium setting, the smallest symmetry group which can support such a Haldane phase is a $Z_2 \times Z_2$ symmetry. However, in the non-equilibrium Floquet setting, it was shown that one can stabilize the same phase, but with an *even smaller* symmetry group (which is proven to be not allowed in equilibrium) – just one Z_2 symmetry (see e.g. Phys. Rev. B 93, (2016); Phys. Rev. X 6, 041001 (2016); Phys. Rev. Lett. 119, 123601 (2017)). Per my understanding, *this* is what is meant by an intrinsically non-equilibrium version of the Haldane SPT phase. On the other hand, the Floquet Hamiltonian that the authors consider explicitly has a $Z_2 \times Z_2$ symmetry, so I am confused why it should be thought of as an intrinsically non-equilibrium FSPT?

Rather than thinking about the system as an intrinsically non-equilibrium FSPT phase, it seems like one might naturally think about it as the fact that H2 exhibits a decoupled edge mode and H1 is basically performing slightly imperfect (with an imperfection set by δ) spin-echo pulses on this edge spin. If this is the case, I might expect the edge oscillations in Fig 2a to decay with a time-scale set by $1/\delta$. Is this observed to be the case numerically?

It would be extremely helpful for the reader if the authors were more clear about the actual circuit that is being run, not just how it was obtained (at the very least in the methods section). For example, this referee found it quite nice that the authors were able to digitally generate 3-body ZXZ terms, but the authors' only comment on how this was done is that "to find the desired quantum circuits, we utilize a neuroevolution method introduced in Ref. [45]..." Reading through ref 45 does not provide any additional insight into this particular ZXZ Hamiltonian. I believe that readers would appreciate a detailed discussion about how the circuits are obtained and any intuition for their design.

The authors plot the z correlation function in Fig 2a, but it would be extremely insightful to see the same level of stability (which is expected) if they prepare the edge spins in a different orientation. For example, does the x correlator exhibit the same long coherence time?

I am wondering if the authors' method for distinguishing between the so-called FSPT vs the thermal phase can actually discriminate between a trivial MBL phase and a thermal phase? For example, I might expect that for very large V there is another regime of MBL behavior? It would be helpful for the reader if the authors could distinguish and independently investigate the thermalization properties and the order parameter.

I'm a bit confused about the authors' discussions surrounding Fig 3b. In particular, it seems that the authors simply prepared cluster states and measured their entanglement and it is not clear how this intersects with the rest of their story?

Moreover, in Fig 3b, I do not know what the energy value index corresponds to? Naively, in the MBL SPT associated with H_2' , I would expect that there would be 2 or 4 fold degeneracy across the entire spectrum owing to localization (and the fact that the authors are preparing cluster states), but from reading the paper, it seems like the authors are looking at a clean system where the topological degeneracy only lives in the ground states? This is a bit confusing given the rest of the MBL story in the manuscript. I also think there are perhaps some incorrect statements which lead to further confusion: for example, in the figure the authors actively mention that the parameters in 3b include additional interactions and finite δ , which I think is incorrect?

The referee really likes figure 3c!

As far as I can tell, Fig 4a presents numerical results rather than experimental data. Curiously, the phase boundary in 4a (at $\delta \sim 0.27$) seems inconsistent with the phase boundary of 4b $\sim 0.1-0.2$. What is the origin of this inconsistency? Is it simply finite size effects

Author Rebuttals to Initial Comments:

List of major changes (marked in red in the main text and the Supplementary Information):

1. We have changed the title to “Observation of Floquet symmetry-protected topological phases with superconducting qubits”.
 2. Following the Referees’ suggestions, we have fabricated a new device and redone the experiments using up to 26 qubits. We have replotted Figs. 1,2,3,4. The corresponding captions are revised accordingly.
 3. We have added two Extended Data Figures (Fig. 5 and Fig. 6) to account for the additional experiments on the two new models with smaller microscopic symmetry groups.
 4. We have added a new section “Further experiments on nonequilibrium SPT phases” in the Methods.
 5. We have added three new sections, Sec. IV “Numerical simulation with noise”, Sec. V “Quantum state tomography”, and Sec. VII “Extended models” in the Supplementary Information.
 6. We have moved some of the figures (about the experimental data from the old superconducting chip) from the main text to the Supplementary Information.
 7. We have revised the abstract, the introduction, and the “Discussion and conclusion” section.
 8. We have added other necessary revisions throughout the whole manuscript to improve the presentation and address the Referees’ comments/suggestions.
-

Response to Referee #1:

We sincerely thank the Referee for their time reviewing the manuscript, accurate summary of the main results, and constructive suggestions. The Referee has provided a long carefully written report, which was invaluable for our efforts to improve the paper. We took the Referee's comments and suggestions very seriously and worked very hard to extend the size of our experimental system from 14 qubits to 26 qubits and to carry out additional experiments to address these comments/suggestions. Thanks to the Referee's report, we have clarified several important points and improved the presentation of the manuscript significantly. The following contains our detailed response to specific points.

Comment 1 of Referee #1: "Over the last couple of years, Floquet Time Crystals, periodically driven systems that break discrete time translation invariance, have been realized in various quantum computing and quantum simulation platforms. Those works have time-crystalline order both in the bulk and at the edge. Crucially, such dynamical phases of matter cannot be realized in equilibrium but only dynamically. In this work, a related concept of a Floquet Symmetry Protected Topological (FSPT) phase has been investigated with a superconducting qubit quantum computer. The essential properties are that time-translational symmetry is broken at the edge of the system, manifesting in periodic modulation of local observables at the edge, while in the bulk such order decays. An essential ingredient to stabilize these phases is many-body localization, which stabilizes the 'order'. In this work circuits are developed that demonstrate a FSPT phase and investigate the edge coherence."

Authors' response: We thank the Referee for their accurate summary of the paper.

Comment 2 of Referee #1: "Over the past couple of years, many theoretical works have studied the concept of FSPT phases and their generalization. This work is essentially an application of these ideas on a quantum computer. As such the novelty of the work is not directly evident. In particular, as the quantum computer works only with 14 qubits, all quantum computations (even including noise models) can be immediately simulated on classical resources. From a theoretical standpoint, therefore these quantum computations are nice to have but they don't help to further advance the field of non-equilibrium quantum phases."

Authors' response: We thank the Referee for raising these crucial points, which have led us to put great effort into extending the size of our experimental system from 14 qubits to 26 qubits, thus dramatically improving the experimental data. With such a large system size, it becomes very challenging to simulate our

experiments on a classical computer. Indeed, taking into consideration experimental imperfections (such as decoherence, pulse distortions, cross talk, etc.), such a simulation requires computing the time evolution of a density matrix of 26 qubits acted upon by about 240 quantum channels. The simulation complexity of this time evolution is similar to the case of simulating a quantum circuit (unitary matrix) of depth 240 acting on 52 qubits. Without deliberately optimized algorithms, an exact simulation of such a large quantum circuit is beyond the capacity of most classical computers. Indeed, as we will mention again below, we were unable to do the exact numerical simulation of experimental data from Fig. 2 taking into account experimental imperfections.

We mention that, in the two most recent related works, one by the Google Quantum AI team and their collaborators (arXiv: 2107.13571v2, recently accepted in Nature) and the other by the Taminiau group and their collaborators (arXiv: 2107.00736, recently accepted in Science), the system sizes reported are 20 and 9, respectively. With the new experimental data, the system size achieved in our experiment is larger than that reported in both Google's and Taminiau's experiments.

We agree with the Referee that many theoretical works have studied the concept of FSPT phases and their generalization over the past couple of years. However, observing such exotic phases in experiments still remains very challenging for state-of-the-art quantum platforms. In our experiment, we explore a fully digital quantum simulation approach, which requires more controllability (and thus is technologically more challenging) compared with the analog approach used in most related previous experiments (see e.g. Refs. [9-11]). An advantage of the digital approach is that it is more versatile for quantum simulation of a much broader class of many-body systems. For instance, in our experiment we have successfully implemented the complex three-body ZXZ interaction, which is not trivial and has not been reported yet in previous related experiments (the Referee #3 also noted this (quote here): "The experiment's ability to faithfully realize a complex three-body ZXZ interaction is quite impressive"). In addition, in our resubmission we used the same approach to carry out additional experiments on two other models; for details, see the manuscript and our reply to Referee #3.

We stress that our work is one of the first experimental demonstrations of such FSPT phases. The only other realization of which we are aware—arXiv:2107.09676v1, which uses Honeywell's trapped ion platform—found an FSPT state with a short coherence time and a system size of ten qubits. Our results, in contrast, show relatively long-lived coherent subharmonic oscillations of the edge spins and cluster stabilizers for a variety of initial states, for a system that is over twice as large, despite the presence of noise and experimental imperfections. Although signatures of discrete time crystals have been reported in many previous works (including the most recent Google and Taminiau experiments), none of these experiments rely on topology as a key ingredient. Unlike the reported discrete time crystals, where spontaneous breaking of the discrete time translational symmetry occurs for local observables throughout the whole system, the FSPT phases observed in our experiment break the time translational symmetry only at the boundaries for topological reasons. The physics studied in our experiment is therefore different from the previous experiments in an essential way. Thus, we think the novelty of our work is justified.

In this revised manuscript, we have redone the experiment with up to 26 qubits and a quantum circuit depth up to 240. To the best of our knowledge, this is the record in reported experiments on simulating nonequilibrium quantum many-body systems in a fully digital way, with such a large system size and circuit depth. We have also carried out additional experiments on two other models to illustrate the general applicability of our digital simulation approach (see the manuscript and our response to Referee #3).

Comment 3 of Referee #1: “The data analysis needs to be improved. Several figure captions don’t describe what kind of error the error bars represent (vertical lines are assumed to represent error bars in Fig. 2). Most likely it is just the s.e.m. over disorder samples. Anyway, instead of me speculating, the authors should clarify what is shown. Moreover, the authors do not perform direct comparisons with exact numerical results. I would find such comparisons helpful in all figures. In particular in Fig. 4b it would be the most natural to have them as the numerical data is even shown in Fig. 4a. Also, the caption of Fig. 4 is therefore misleading. It should rather say “Numerical phase diagram ...” to avoid confusion.”

Authors’ response: We thank the Referee for these constructive suggestions. Yes, the error bars represent the standard error of the mean (s.e.m.) over disorder samples. In the revised manuscript, we have clarified this for all figures.

We agree with the Referee that a direct comparison between experimental and exact numerical results is very helpful. We have added such a comparison to Fig. 3, Fig. 4, and the Extended Data Fig. 5 and Fig. 6, where exact numerical simulations with experimental imperfections are feasible. For Fig. 2, since we have increased the system size to 26 qubits, an exact numerical simulation becomes very challenging (as explained above) and we are not able to do such a simulation due to limited classical computational resources.

In addition to adding the numerical comparison to Fig. 4b, we also followed the Referee’s suggestion and changed the title of the caption for Fig. 4 to “Numerical phase diagram and experimental detection of the phase transition.” to avoid confusion. We also added the words “Numerical simulation” explicitly to Fig. 4a and revised the caption accordingly.

Comment 4 of Referee #1: “One has to stare for quite a while on Fig. 1c to understand what the authors intent to show. This could be improved.”

Authors’ response: We thank the Referee for this helpful suggestion. We agree that the original Fig. 1c was confusing. In the revised manuscript, we have replotted Fig. 1c to show the two-fold degeneracy of the quasienergy spectrum of U_F in a more straightforward manner. We also improved the caption accordingly to avoid possible confusions.

Comment 5 of Referee #1: “For Floquet protocols, one typically expects measurements after every completed period. It is not done in that way in Fig. 2a. This should be clearly discussed in the text to clarify the situation (perhaps the authors want to report and show only full periods?). Also x-axis ticks and labels on panel 2a would be useful.”

Authors’ response: We thank the Referee for these valuable suggestions. The reason why we originally measured σ_j^z at different time steps within a period is that we wanted to see in more detail how this quantity evolves. In fact, doing measurements only after every completed period would simplify our experiment. In the revised version, we followed the Referee’s suggestions and carried out a new experiment with 26 qubits. We measured σ_j^z after every half period (so that we can see how $\overline{\sigma_j^z}$ evolves under H_1 and H_2 , respectively) and replotted Fig. 2a with x-axis ticks up to 40 driving cycles.

Comment 6 of Referee #1: “Fig. 3b shows the entanglement spectrum of a reduced density matrix directly obtained after state preparation of a cluster state. While some of the eigenvalues are somewhat nearby, degeneracies are absent and the spectrum is rather different from what is expected theoretically. Given the low depth of the circuits it should be critically discussed where this discrepancy comes from (currently only a vague sentence is included in the main text). This could help to understand the limitations of their system. It would be also important to state how the authors extract the density matrix from the measurements. The subsystem is so small that it should be possible to learn something about the device from these imperfections.”

Authors’ response: We thank the Referee for raising these important points, which were valuable in helping us improve the manuscript.

In Fig. 3b, we first prepare a random eigenstate of H_2 with $V_k = h_k = 0$, evolve the state with $H(t)$ for one period (this time V_k and h_k are not zero), and then do full quantum state tomography to obtain the reduced density matrix and calculate the entanglement spectrum. Here, the reduced density matrix is *not* directly obtained after the state preparation, but after one period of time evolution under $H(t)$. The aim of doing this is to extract useful information about the eigenvalues of U_F : the degeneracy of the evolved states indicates that the eigenstates of U_F have a large overlap with the corresponding initially prepared cluster states, which is also consistent with the topological properties of U_F .

We believe there are two factors that lead to the deviation of our results from the expected degeneracies: 1) the unavoidable experimental imperfections, such as decoherence, pulse distortions, and crosstalk effect; 2) finite size effects. Indeed, in the experiment shown in Fig. 3b, the system size in the original manuscript was $L = 6$, which is very small and lead to non-negligible coupling between the edge modes.

In the revised manuscript, we have redone the experiment with $L = 10$ qubits. Now, the degeneracy of the entanglement spectrum is much clearer, which also agrees with numerical simulations taken into account the experimental imperfections. More specifically, the experimental imperfections considered in our numerical simulations include the two-qubit gate infidelity and slight measurement probability uncertainty, which is around 0.0055 (0.0095) when measuring the qubit population in the ground (excited) state. We have also added a new Sec. V in the Supplementary Information to show in detail how the density matrix is obtained in our experiments.

Comment 7 of Referee #1: “Logarithmic entanglement growth is claimed to be present in the exact numerical results. Looking at Fig. S1c, I wonder where authors see logarithmic growth?”

Authors’ response: We thank the Referee for noting this point. In a many-body localized (MBL) system, the entanglement entropy is expected to grow logarithmically (see e.g. PRL 109, 017202 (2012); PRL 110, 260601 (2013)). Deep in the FSPT phase, our system is many-body localized, and thus we expect logarithmic entanglement growth. In Fig. S1c, for the chosen parameters $J = \Delta J = 1$, $h = \Delta h = V = \Delta V = \delta = 0.05$, our model is indeed in the FSPT phase (and is thus many-body localized), but the localization length is not very small since the disorder strength is not large enough. Consequently, we can see logarithmic growth only after a very long evolution time. Indeed, in the original figure, the logarithmic entanglement growth

started to appear only after about 20 periods.

In the revised manuscript, we have redone the calculation with larger disorder strength (the parameters are chosen as: $J = 1$, $\Delta J = 4$, $\hbar = \Delta\hbar = V = \Delta V = \delta = 0.05$). This time, the localization length is smaller, and the logarithmic entanglement growth is more evident. We have replotted Fig. S1 with the new data and explicitly added a logarithmic fit to Fig. S1c. We also revised the caption accordingly.

Comment 8 of Referee #1: “The conclusions of the work discuss the realization of the non-equilibrium phase with nontrivial topological properties. In the last paragraph, the authors state that ‘The controllability and scalability of the superconducting platform demonstrated in our experiment. . .’. I don’t think that this work is a strong showcase of scalability of superconducting qubits. I suggest to clarify this statement or remove it. Currently it does not carry any weight.”

Authors’ response: We thank the Referee for this helpful suggestion. In the revised manuscript, we followed the Referee’s suggestion and deleted this statement.

Comment 9 of Referee #1: “The theoretical concept of dynamical phases without any equilibrium analogue and with nontrivial topological properties is very interesting. These concepts have been introduced theoretically in previous works. I am not sure what this study can add to the current understanding. Somewhat open question in the field could be late time/large system effects, and certainly behavior in two dimensions etc. The current work is a nice demonstration on a quantum computer, but all results could be easily simulated on a classical computer. Therefore, this work does not seem to fulfill the stringent requirements of Nature but seems to be rather suited for a more specialized journal.”

Authors’ response: We thank the Referee for pointing out that the “theoretical concept of dynamical phases without any equilibrium analogue and with nontrivial topological properties is very interesting”. We hold the same opinion as the Referee on this point, and this is exactly the motivation for us to carry out experiments to show signatures of such phases in the laboratory. As discussed above, in this revised manuscript we have fabricated a new superconducting chip and extended the experiment to a system size as large as 26 qubits, with a circuit depth up to 240. With such a large system size and circuit depth, it becomes very challenging in simulating our experiments with a classical computer. We agree with the referee that some important questions, such as late time effects and behavior in higher dimensions, still remain open. Without a doubt, these are exceptionally challenging problems and solving them completely requires further breakthroughs in both theory and quantum technologies. Our work is one of the first two experimental demonstrations of exotic FSPT phases. With the newly added results, we have achieved a full digital simulation of FSPT phases with the largest system size reported in the literature. We thus believe that this work indeed represents a significant step in studying nonequilibrium topological phases and would have broad impact.

In summary, we greatly appreciate the Referee’s valuable comments/suggestions, which were very helpful for us in improving the manuscript. Following these comments and suggestions, we have carried out additional experiments with 26 qubits (as opposed to 14 qubits in the original submission) on a newly fabricated superconducting device. We have carefully addressed all the important points raised by the Referee and significantly improved the presentation of our results. We hope that this substantially improved manuscript

will satisfy the Referee and convince them to recommend publication of this work in Nature.

Response to Referee #2:

We thank the Referee for their careful reading of the manuscript and valuable comments/suggestions, which have helped us improve the paper significantly. We particularly appreciate the Referee's very positive evaluation that our work "presents a compelling demonstration of FSPT phase of matter". Based on their report, we have clarified several crucial points and improved the presentation significantly. Now, the paper is substantially strengthened and we believe is suitable for publication in Nature. The detailed response to the Referee's comments is provided below.

Comment 1 of Referee #2: "The manuscript entitled "Observation of a Floquet symmetry-protected topological phase with superconducting qubits" by Zhang et al. presents a compelling demonstration of Floquet symmetry-protected topological (FSTP) phase of matter implement digitally using a 1D array of superconducting qubits.

The main results of the manuscript are, first, the observation of the FSPT phase and the 2T periodicity of the edge qubits (1, 14) with essentially no net magnetization of the bulk qubits (2-13). This was observed for randomized input states. Second, the authors tested the time-dependence of local stabilizers, again showing 2T-periodicity at the edges, with T-periodicity in the bulk. Lastly, the authors demonstrate a transition from the TSPT phase to a thermal phase, realized by increasing the Hamiltonian parameters that reduced many-body localization. As I describe below, I have a few suggestions on this part.

Overall, I found the manuscript straightforward to read and follow. The experiments are carefully documented and support the statements in the paper. I think the topic will be of interest to physicists and quantum engineers alike. Whether this belongs in Nature or Nature Physics is an editorial decision."

Authors' response: We thank the Referee for the nice summary. We appreciate the Referee's judgement that our work "presents a compelling demonstration of FSPT phase" and that "the topic will be of interest to physicists and quantum engineers alike". In the revised manuscript, we have extended our experiments to 26 qubits and carried out additional experiments on two other models to illustrate the general applicability of our digital simulation approach. The revised manuscript is now substantially stronger than the original version.

Comment 2 of Referee #2: "(1) In Fig. 1, there are green qubits and yellow qubits. The green qubits are the 14 used in this experiment. The yellow ones are unused qubits. The authors should mention the meaning of the yellow color scheme in the caption (they already mention the green). Also, not all lattice sites have yellow dots. Does this mean those qubits did not work?"

Authors' response: We thank the Referee for raising this import point. Yes, the Referee is correct that the green qubits denoted the ones used in the experiment, and the yellow ones are functional qubits that were not selected for the experiment due to limited gate fidelity. The other lattice sites marked in red are unfunctional. For the revised manuscript, we have fabricated a new chip and have redone the experiment with up to 26

qubits. We have replaced Fig. 1e with a new image and revised the caption of Fig. 1 accordingly.

Comment 3 of Referee #2: “(2) The authors set $T = 2T_1 = 2$. I may have missed it, but it would be good to specify how long this is in ns or μ s. Also, perhaps use something to distinguish this T_1 from relaxation T_1 , e.g., t_1 .”

Authors’ response: We thank the Referee for this helpful suggestion. In our experiment, we use a quantum circuit to digitally simulate the dynamics of the system under $H(t)$. The time for implementing the single-qubit gates $X(\theta)$ and $Y(\theta)$ is 50 ns, and the average time for implementing the controlled-Z gate is about 40 ns. It takes around 11.6 μ s to run the circuit of a depth up to 240, with 1.2 μ s for measurement. The average energy relaxation time T_1 is above 30 μ s. We have clarified this important point in the revised manuscript (see the second sentence below Eq. (1) in the main text). We replaced T_1 in defining the Hamiltonian with T^l to distinguish it from the relaxation time T_1 .

Comment 4 of Referee #2: “(3) The authors talk about disorder averaging at the beginning of the Symmetry breaking section, but only explain that it means randomizing the input states at the beginning of the next section (Localization-protected topological states). I would state earlier what it means.”

Authors’ response: We thank the Referee for this helpful suggestion. In our experiment, we considered both random disorder realizations of $H(t)$ (by choosing J_k , V_k , and h_k randomly) and random initial states. For each random disorder realization, we typically prepare only one random initial state to save experimental efforts. Our results are averaged over disorder realizations. In the revised manuscript, we follow the referee’s suggestion and have clarified this at the beginning of the “Symmetry breaking at boundaries” section.

Comment 5 of Referee #2: “(4) What is meant by “external circuit errors” and what is meant by “thermalization”? I would explain this terminology in a sentence or so.”

Authors’ response: By “external circuit errors”, we mean the experimental imperfections (such as decoherence, pulse distortions, cross-talk effect, etc.) which will reduce the gate fidelity and lead to the decaying magnetization envelope eventually. By “internal thermalization”, we mean the thermalization due to the model itself. Both external circuit errors and internal thermalization would result in the decay of the edge magnetization. In order to establish the observation of an FSPT phase, we need to make sure that the decay of magnetization for the edge spins is due to experimental imperfections rather than internal thermalization. This is verified by the echo circuit, as shown in Fig. 2f. In the revised manuscript, at the end of the section “Symmetry breaking at boundaries”, we have explicitly explained the meaning of external circuit errors and internal thermalization.

Comment 6 of Referee #2: “(5) In the Localization-protected topological states section, the authors say that the periodic oscillations of the edge (Fig3c) are in “sharp contrast to the dynamics of bulk magnetization, which decay rapidly to zero”. I did not see this so obviously in Fig. 3c. It may be worthwhile to take line cuts to make it clear. If I understood correctly, the dark red at short times becomes whiter toward $t/T = 5$, but it is still seems quite red to me (at least on my computer). Maybe change the wording, or show how it is

decaying, alter the color scale, etc.”

Authors’ response: We thank the Referee for raising this point. Fig. 3c plots the time-dependence of local stabilizers, rather than magnetization, in the FSPT phase. The stabilizers at the boundaries oscillate with a $2T$ periodicity, whereas the stabilizers in the bulk oscillate with a T periodicity. In the ideal situation, the bulk stabilizers will exhibit persistent oscillations even in the infinite-time limit due to their finite overlaps with local integrals of motion. This is different from the dynamics of the bulk magnetization, which decays rapidly to zero. In the revised manuscript, we have replotted Fig. 3 with a larger system size and clarified this point (see the last paragraph of Sec. “Localization-protected topological states” in the main text).

Comment 7 of Referee #2: “(6) I found the phase transition to be the least supported part of the manuscript. I have a few suggestions here. First off, make it clear that 4a is a numerical simulation in the caption. It is called numerical simulation in the main text. Then, I would suggest explaining why the phase boundary is not so sharp. You indicate it is because of finite size effects. Is there anything more to say about it to help the reader understand? Lastly, if I understand correctly, Fig 4b corresponds to $V = 0$ in Fig 4a. If that is indeed the case, could you put a line cut from 4a in to 4b, so we can see how well it matches?”

Authors’ response: We thank the Referee for these valuable suggestions. In the revised manuscript, we followed these suggestions and have redone both the numerical simulation and the experiment with more disorder instances. We have changed the title of the caption for Fig. 4 to “Numerical phase diagram and experimental detection of the phase transition” and added the words “Numerical simulation” explicitly to Fig. 4a to make it clear that Fig. 4a is a numerical simulation.

The Referee is correct that Fig. 4b corresponds to $V = 0$ in Fig. 4a. We replotted Fig. 4b with our new experimental data. Following the Referee’s suggestion, we added numerical comparisons and a purple vertical dashed line in the inset of Fig. 4b to indicate the numerically determined phase boundary from Fig. 4a. With the new device and additional experiments where we average over more disorder instances, the phase transition point is estimated to be at $\delta = 0.3$, which is close to the theoretical prediction ($\delta_c = 0.34$) from Fig. 4a. We revised the caption of Fig. 4 accordingly.

We believe that the reason why the phase boundary in Fig. 4a is not so sharp is mainly due to finite size effects. Theoretically, if the system size is too small, then the coupling between the two edge modes is not negligible, which will decrease the central subharmonic peak height in the FSPT phase. In order to obtain this phase diagram, the numerical computational effort is huge, and we are restricted to a small system size $L=8$ due to limited classical computational resources. In fact, to numerically locate a sharp phase boundary still remains a notorious challenge in the MBL literature (see e.g., Rev. Mod. Phys. 91, 021001 (2019)). In the revised manuscript, we have added a sentence “for a small system size $L = 8$, the coupling between the two edge modes is not negligible and thus will decrease the central subharmonic peak height and result in a blurred boundary” at the end of the first paragraph of the section “Phase transition” in the main text, to explain why a small system size will make the phase boundary not so sharp.

Comment 8 of Referee #2: “(7) In the Methods, you stated that you could not observe the logarithmic entanglement growth. This is due to “decoherence and other experimental imperfections”. What are the experimental imperfections? I would be clear here. Do you mean gate fidelity is not high enough? Do you

mean Trotterization and the available circuit depth don't allow for it? Do you mean that characterizing entanglement via tomography scales very poorly with system size? All of the above? Do simulations using your decoherence and your experimental imperfections (what are they?) indicate that you do not expect to see the entanglement growth? The SM does not really say much more about this, other than we need more efficient ways to measure the entanglement growth. I would expand the discussion a bit, certainly beyond "decoherence and other experimental imperfections".

Authors' response: We thank the Referee for this helpful suggestion. There are two key factors that prevent us from observing a logarithmic entanglement growth in our experiment: 1) experimental imperfections, such as decoherence, crosstalk, and pulse distortions, which all result in a limited gate fidelity; 2) the small system size used in our experiment to measure the entanglement growth. In our experiment, we utilize a full tomography method, whose complexity grows exponentially with the system size, to obtain the entanglement entropy. Since we need to do full tomography at different time points and for many different random disorder instances, the required experimental effort is huge and we are limited to $L = 6$ qubits. For such a small system size, the logarithmic entanglement growth is not very evident even in the numerical simulation. We clarify that we do not do Trotterization in our experiment, since this will introduce a Trotter error and increase the circuit depth in simulating U_F .

In the revised manuscript, we have clarified what the experimental imperfections are and explained in more depth why we cannot observe a logarithmic entanglement growth in our experiment (see the section "Logarithmic entanglement growth" of Methods). We also expanded the discussion in Sec. VI of the Supplementary Information. Finally, we numerically simulated the experiment with experimental noise taken into account and added the result to Fig. S9b.

Comment 9 of Referee #2: "(8) I would suggest the authors not use overused idioms like "paves the way" or "opens the door". Can you think of more precise terms for what your work actually enables?"

Authors' response: We thank the Referee for this constructive suggestion. In the revised manuscript, we have replaced the sentence "Our work paves the way..." with "Our results establish a versatile digital simulation approach to exploring exotic non-equilibrium phases of matter with current noisy intermediate-scale quantum processors", at the end of the abstract. We also deleted the sentence "This work opens the door to harnessing this exotic phase of matter for practical quantum information processing" in the introduction section. In addition, we added a sentence "The digital quantum simulation approach explored in our experiment is generally applicable to simulating a wide range of nonequilibrium systems hosting unconventional topological phases, including those with multi-body interactions.", at the end of the "Discussion and conclusion" section.

In summary, we greatly appreciate the Referee's positive evaluation of our work and their very helpful comments/suggestions. Following these comments and suggestions, we have made substantial revisions to the manuscript and improved the presentation significantly. We hope that this substantially improved manuscript will satisfy the Referee and convince them to recommend acceptance of this work in Nature.

Response to Referee #3:

We thank the Referee for their time reviewing the manuscript and appreciate their insightful comments/suggestions, which have led us to carry out additional experiments on two new models with only Z_2 or even no microscopic symmetries. Based on the Referee's report, we have clarified several crucial points and improved the presentation significantly. Now the paper is significantly strengthened and we believe is suitable for publication in Nature. The detailed response to the Referee's comments is provided below.

Comment 1 of Referee #3: “The manuscript by Zhang et al. describes the realization of a Floquet symmetry protected topological phase using a one dimensional chain ($L = 14$) of superconducting transmon qubits. The manuscript builds upon the recent excitement surrounding the discovery that there exist intrinsically non-equilibrium phases of matter that do not have an equilibrium counterpart. The experiment's ability to faithfully realize a complex three-body ZXZ interaction is quite impressive, although very few details are given regarding how this is done. Of course, as the authors' direct comparison to numerics already indicates, a 14 qubit quantum simulation can readily be done on a classical computer, so this itself is perhaps not the defining feature; on the other hand, I do think that the experimental demonstration of the underlying conceptual framework surrounding Floquet SPT phases (as well as the realization of complex many-body interactions) is something that has the potential to have wide-ranging impacts. However, to be totally honest, despite multiple readings of the manuscript, I remain confused about a number of key points and would very much appreciate the authors' explanations.”

Authors' response: We thank the Referee for the nice summary and greatly appreciate their positive evaluation that “the experimental demonstration of the underlying conceptual framework surrounding Floquet SPT phases (as well as the realization of complex many-body interactions) is something that has the potential to have wide-ranging impacts”. We also thank the Referee for pointing out that “faithfully realiz[ing] a complex three-body ZXZ interaction is quite impressive”. We agree with the Referee on this point.

For the revised manuscript, we have fabricated a new superconducting chip and extended our experiment to a system size as large as 26 qubits and a circuit depth up to 240. With such a large system size and circuit depth, it becomes very challenging to simulate our experiments with experimental noise on a classical computer (see our response to Comment 2 of Referee #1 for a detailed discussion). We carried out additional experiments on two new models, one with only a Z_2 symmetry and the other with no microscopic symmetry, and observed persistent edge oscillations protected by Z_2 and time-translation symmetries in the former case, and by an emergent dynamical symmetry in the latter case (as discussed in depth below). We added more details about how we obtained the analytical quantum circuits for simulating the three-body ZXZ interaction (see Supplementary Information Sec.III A) and improved the presentation significantly.

Comment 2 of Referee #3: “Perhaps most importantly, the authors emphasize that they are reporting the observation of “an intrinsically non-equilibrium FSPT phase.” I am a bit confused whether this is actually correct. This question may get a bit long, but I want to be absolutely sure that I am being as clear as possible. In particular, the physics that the author's are exploring can be understood by analogy to the equilibrium AKLT/Haldane spin chain, which exhibits a decoupled spin-1/2 edge degree of freedom. In the equilibrium setting, the smallest symmetry group which can support such a Haldane phase is a $Z_2 \times Z_2$ symmetry. However, in the non-equilibrium Floquet setting, it was shown that one can stabilize the same phase, but with an *even smaller* symmetry group (which is proven to be not allowed in equilibrium) – just one Z_2

symmetry (see e.g. Phys. Rev. B 93, (2016); Phys. Rev. X 6, 041001 (2016); Phys. Rev. Lett. 119, 123601 (2017)). Per my understanding, *this* is what is meant by an intrinsically non-equilibrium version of the Haldane SPT phase. On the other hand, the Floquet Hamiltonian that the authors consider explicitly has a $Z_2 \times Z_2$ symmetry, so I am confused why it should be thought of as an intrinsically non-equilibrium FSPT? ”

Authors’ response: We thank the Referee for this insightful comment, which has led us to carry out additional experiments on two new models with only Z_2 or even no microscopic symmetry.

We agree with the Referee that our Floquet Hamiltonian $H(t)$ indeed has a $Z_2 \times Z_2$ symmetry, which can support the AKLT/Haldane SPT phase even in the equilibrium setting. However, we argue that it is still intrinsically non-equilibrium in the sense that we cannot have an effective local time-independent Hamiltonian that features spontaneous breaking of time-translational symmetry at the system’s boundaries; if we were to calculate $i \log(U_F)$, the effective Hamiltonian obtained would be nonlocal. Indeed, a no-go theorem has been proven in the time crystal literature that quantum space-time crystals in equilibrium are not possible (see Watanabe & Oshikawa, PRL 114, 251603 (2015)). Similar considerations hold for FSPT phases regardless of their symmetry group, since these phases feature time-translation symmetry breaking on the boundaries that is robust to symmetric perturbations (see our response to Comment 3 below). The Referee is correct that in the non-equilibrium Floquet setting, even a smaller symmetry group would be enough to stabilize the AKLT/Haldane phase, as shown in the renowned papers mentioned by the Referee. The Referee’s definition of “intrinsically non-equilibrium” is more about stabilizing a topological phase with smaller symmetry group (which is not possible in equilibrium), whereas our definition is more about spontaneous breaking of time-translational symmetry. In order to avoid possible confusions, we have removed the word “intrinsically” throughout the paper.

Motivated by the Referee’s comment, we have carried out additional experiments on two new models, one with only Z_2 symmetry (the periodically driven random Ising model) mentioned by the Referee and the other (the quasi-periodically driven model hosting an emergent dynamical SPT phase) with no microscopic symmetry. We note that a pioneering experimental demonstration of the quasi-periodically driven model with trapped ions has been reported recently (arXiv:2107.09676). For both models, we have observed robust subharmonic edge oscillations (see the newly added Extended Data Fig. 5 and Fig. 6, and Sec. VII of the Supplementary Information for details). With these new results, we have unambiguously demonstrated the observation of intrinsically non-equilibrium (according to the Referee’s definition) SPT phases.

Comment 3 of Referee #3: “Rather than thinking about the system as an intrinsically non-equilibrium FSPT phase, it seems like one might naturally think about it as the fact that H_2 exhibits a decoupled edge mode and H_1 is basically performing slightly imperfect (with an imperfection set by δ) spin-echo pulses on this edge spin. If this is the case, I might expect the edge oscillations in Fig 2a to decay with a time-scale set by $1/\delta$. Is this observed to be the case numerically?”

Authors’ response: We thank the Referee for raising this important question, which has helped us improve the presentation and clarify the misunderstanding. It is true that H_2 exhibits decoupled edge modes, but the dynamics of these edge spins cannot be simply understood from spin-echo pulses on the edge spins. In fact, as discussed in Sec. I of the Supplementary Information, every eigenstate of U_F is two-fold degenerate and has a cousin separated by quasi-energy π . This is distinct from H_2 whose eigenstates are four-fold degenerate and whose edge modes—in the cluster limit ($V_k = h_k = 0$)—can be regarded as fully decoupled

from the system (thus leading to the four-fold degeneracy). In the Floquet setting, the eigenstates of U_F are degenerate cat-like states, and thus the edge spins at the two ends are correlated to each other. In this case, one cannot regard the edge modes as fully decoupled spins, and the dynamics of the edge spins involves many-body effects. This is why the edge oscillations are robust to perturbations that respect the $Z_X Z_Z$ symmetry, including nonzero δ .

Based on the Referee’s question, we have carried out numerical simulations with varying δ . Our results are shown in the following Response Figure 1. From this figure, it is clear that the edge oscillations persist and do not decay with increasing δ . This explicitly shows that the dynamics of the edge spins involves many-body effects and cannot be simply understood from spin-echo pulses on the edge spins. A similar “subharmonic rigidity” is found in time crystals, which are another class of intrinsically nonequilibrium phases. In the revised manuscript, we stressed the robustness to nonzero δ (see the first paragraph of section “Symmetry breaking at boundaries” in the main text).

Response Figure 1: Numerical simulation of the dynamics of $\overline{\langle \sigma_1^z \rangle}$ averaged over 500 random disorder instances. Here, the initial states are chosen as random product states in the z basis. The parameters are chosen as $J = \Delta_J = 1$, $V = \Delta_V = \hbar = \Delta_h = \delta = 0.05$, and the lattice size is $L = 100$. This simulation is implemented with the TEBD algorithm with time step $\Delta t = 0.1$ and cutoff error smaller than 10^{-8} .

Comment 4 of Referee #3: “It would be extremely helpful for the reader if the authors were more clear about the actual circuit that is being run, not just how it was obtained (at the very least in the methods section). For example, this Referee found it quite nice that the authors were able to digitally generate 3-body ZXZ terms, but the authors’ only comment on how this was done is that “to find the desired quantum circuits, we utilize a neuroevolution method introduced in Ref. [45]. . .” Reading through ref 45 does not provide any additional insight into this particular ZXZ Hamiltonian. I believe that readers would appreciate a detailed discussion about how the circuits are obtained and any intuition for their design.”

Authors’ response: We thank the Referee for this constructive suggestion. We agree with the Referee that it is quite nice to be able to digitally generate 3-body ZXZ terms in an analytical fashion, even under the

condition that the coupling strengths are random. To obtain a practical circuit to simulate H_2 , we exploit a neuroevolution algorithm introduced in one of our previous papers (Ref. [45]), which maintains a salient performance in quantum circuit architecture design, to output a near optimal quantum circuit. We have now added a new section (Sec.III A) in the Supplementary Information to explain the details of this algorithm for the Referee and other readers' interest. In short, the analytical circuits for simulating the ZXZ terms are obtained by removing all gates that are close to the identity and merging nearby gates for the output circuits from the neuroevolution algorithm.

Following the Referee's suggestion, in the revised manuscript, we have explicitly shown the actual circuit to digitally generate the 3-body ZXZ terms in Fig. S4, and added more details to Sec. III A of the Supplementary Information about how we obtain the circuits used in our experiment.

Comment 5 of Referee #3: "The authors plot the z correlation function in Fig 2a, but it would be extremely insightful to see the same level of stability (which is expected) if they prepare the edge spins in a different orientation. For example, does the x correlator exhibit the same long coherence time?"

Authors' response: If we prepare the edge spins in the x direction and measure the dynamics of $\langle \sigma_k^x \rangle$, it will decay to zero quickly, as shown in the following Response Figure 2.

Response Figure 2: Numerical simulation of the dynamics of $\overline{\langle \sigma_k^x \rangle}$ averaged over 200 random disorder instances. Here, the initial states are chosen as random product states along the x direction. The parameters are chosen as $J = \Delta_J = 1$, $V = \Delta_V = \hbar = \Delta_h = \delta = 0.05$, and the lattice size is $L = 26$. This simulation is implemented with the TEBD algorithm with time step $\Delta t = 0.1$ and cutoff error smaller than 10^{-8} .

We can understand the quick decay of $\langle \sigma_1^x \rangle$ by examining H_2 in the idealized cluster limit ($V_k = h_k = 0$): the edge operators for the decoupled left edge mode that behaves as a free spin-1/2 particle are: $\Sigma_1^x = \sigma_1^x \sigma_2^z$, $\Sigma_1^y = \sigma_1^y \sigma_2^z$, and $\Sigma_1^z = \sigma_1^z$. The x component of the left edge mode in fact involves σ_2^z , which leads to the decay of $\langle \sigma_1^x \rangle$. A simpler way of understanding this follows from noting that σ_1^x does not commute with H_2 .

This is different from the case of $\langle \sigma_1^z \rangle$, where σ_1^z commutes with H_2 .

In the revised manuscript, we have clarified this important point (see the newly added last paragraph of Sec.I.B.3 of the Supplementary Information).

Comment 6 of Referee #3: “I am wondering if the authors’ method for distinguishing between the so-called FSPT vs the thermal phase can actually discriminate between a trivial MBL phase and a thermal phase? For example, I might expect that for very large V there is another regime of MBL behavior? It would be helpful for the reader if the authors could distinguish and independently investigate the thermalization properties and the order parameter.”

Authors’ response: We thank the Referee for raising these insightful questions. In this paper, we use the variance of the subharmonic spectral peak height to probe the transition point. For both the trivial MBL phase and the thermal phase, there is no breaking of discrete time translational symmetry, and hence the subharmonic spectral peak is absent. As a result, our method cannot distinguish these two phases. We agree with the Referee that it is worthwhile to study the thermalization/MBL properties for very large V . The topological and trivial MBL phases can be distinguished, e.g., by studying Edwards-Anderson-like string order parameters. However, since our paper focuses on distinguishing the FSPT and thermal phases, this question is somewhat tangential to the main topic. We thus would like to leave it for future studies.

Following the referee’s comment, in the revised manuscript we have added a paragraph (see the last paragraph of Sec.I.B.4 of the Supplementary Information) to discuss this point.

Comment 7 of Referee #3: “I’m a bit confused about the authors’ discussions surrounding Fig 3b. In particular, it seems that the authors simply prepared cluster states and measured their entanglement and it is not clear how this intersects with the rest of their story?”

Authors’ response: We thank the Referee for raising this question, which reflects that we probably did not explain sufficiently clearly the motivation for showing Fig. 3b and how we obtained this figure. In Fig. 3b, we first prepare a random eigenstate of H_2 with $V_k = h_k = 0$, evolve the state with $H(t)$ for one period (this time V_k and h_k are not zero), and then do a full quantum state tomography to obtain the reduced density matrix and calculate the entanglement spectrum. Here, the reduced density matrix is *not* directly obtained after the state preparation, but after one period of time evolution under $H(t)$. Our motivation for plotting Fig. 3b is to show the topological nature of U_F . In experiment, it is impractical to prepare the system in an exact eigenstate of U_F . Thus, we first prepare a random cluster state, which has a large overlap with one of the eigenstates of U_F , and then evolve the state for one period to illustrate the topological feature of U_F .

In the revised manuscript, we have redone the experiment with 10 qubits (the original manuscript used only 6 qubits). With the new device, the two- and four-fold degeneracy of the entanglement spectrum for open and periodic boundary conditions respectively is more evident. We have also clarified further the motivation for showing Fig. 3b (see the second paragraph of section “Localization-protected topological states” in the main text) and how we obtained this figure (see Supplementary Information Sec. V).

Comment 8 of Referee #3: “Moreover, in Fig 3b, I do not know what the energy value index corresponds

to? Naively, in the MBL SPT associated with H_2' , I would expect that there would be 2 or 4 fold degeneracy across the entire spectrum owing to localization (and the fact that the authors are preparing cluster states), but from reading the paper, it seems like the authors are looking at a clean system where the topological degeneracy only lives in the ground states? This is a bit confusing given the rest of the MBL story in the manuscript. I also think there are perhaps some incorrect statements which lead to further confusion: for example, in the figure the authors actively mention that the parameters in 3b include additional interactions and finite δ , which I think is incorrect?"

Authors' response: In order to obtain the entanglement spectrum, we first obtain the reduced density matrix ρ_{half} of half of the system. We then diagonalize $-\log \rho_{\text{half}}$ to obtain the entanglement spectrum. The energy value index labels the eigenvalues of $-\log \rho_{\text{half}}$. For instance, the energy value index 1 corresponds to the smallest eigenvalues of $-\log \rho_{\text{half}}$.

The Referee is correct that, in the cluster-state limit, there would be 2 and 4 fold degeneracy (for the open and periodic boundary conditions, respectively) across the entire spectrum. In our experiment, what we prepare in Fig. 3b are random cluster states, which are in general highly-excited eigenstates (rather than ground states) of H_2 with $V_k = h_k = 0$, which are then evolved over one period by U_F . The resulting states are no longer eigenstates of H_2 . In the revised manuscript, we no longer use the notation H_2' to improve the presentation.

The parameters in Fig. 3b indeed include additional interactions (nonzero V_k and h_k) and finite $\delta = 0.1$. As explained above, Fig. 3b actually shows the entanglement spectra of the states after one driving period under $H(t)$ with nonzero V_k , h_k , and δ .

The Referee's question on this point reflects the insufficient clarity in our original presentation. In the revised manuscript, we have clarified the meaning of "energy value index" in the caption of Fig. 3b, and that the entanglement spectrum in Fig. 3b is obtained after one driving period (see the caption of Fig. 3b and the second paragraph of section "Localization-protected topological states" in the main text). We improved the presentation accordingly.

Comment 9 of Referee #3: "The Referee really likes figure 3c!"

Authors' response: We greatly appreciate the Referee's positive evaluation of Fig. 3c. With the newly fabricated chip, we have repeated the experiment with a larger system size (10 qubits, in contrast to 6 qubits in the original submission).

Comment 10 of Referee #3: "As far as I can tell, Fig 4a presents numerical results rather than experimental data. Curiously, the phase boundary in 4a (at delta 0.27) seems inconsistent with the phase boundary of 4b 0.1-0.2. What is the origin of this inconsistency? Is it simply finite size effects?"

Authors' response: We thank the Referee for raising this important question. We believe there are four major reasons for the inconsistency.

First, the phase boundary calculated in Fig. 4a is not accurate due to finite size effects and limited number

of disorder instances. In computing this phase diagram, we discretize the δ - V plane into a 30×20 grid, and for each point of the grid we simulate the dynamics of the system up to 60 cycles using exact diagonalization. We do this for 1000 disorder instances, and then do a Fourier transformation to obtain the central subharmonic peak. Similar to many works on MBL, the required numerical computational effort is huge. Thus, we are limited to a small system size $L = 8$, and the computed boundary is not very accurate.

Second, although our numerical simulation is limited to $L = 8$, our experiment in the original submission used 14 qubits. In other words, the lattice sizes for the numerical simulation and experiment were different.

Third, due to various inevitable experimental imperfections, such as decoherence, pulse distortions, and cross-talk effects, our gate fidelity is limited.

Fourth, in our experiment, we only implemented 20 disorder instances to save experimental efforts. This leads to an inaccuracy in locating the phase transition point and also contributes to the inconsistency.

In the revised manuscript, we have redone the numerical simulation with longer time evolution and more disorder instances. For a better comparison between our experimental result and numerical prediction with the same system size, we also repeated the experiment with $L = 8$ and *more* disorder realizations (increased the number from 20 in the original submission to 50 in the current version). Now, the experimentally estimated phase transition point is around $\delta \sim 0.3$, which is close to that obtained from numerical simulation (~ 0.34). In addition, we also clarified the possible resources for the remaining 0.04 inconsistency (see the last paragraph of Sec. “Phase transition” of the main text).

In summary, we greatly appreciate the Referee’s invaluable comments/suggestions. Following these comments and suggestions, we have carried out additional experiments to show the observation of intrinsically nonequilibrium FSPT phases. We made substantial revisions to the manuscript, clarified all the confusing points, and improved the presentation significantly. We hope that this noticeably improved manuscript will satisfy the Referee and convince them to recommend acceptance of this work in Nature.

Reviewer Reports on the First Revision:

Referees' comments:

Referee #1 (Remarks to the Author):

The authors have carefully addressed and implemented the suggestions of the referees. I think the manuscript is clearly written now and I don't have any further specific suggestions to the authors.

Referee #2 (Remarks to the Author):

I'd like to thank the authors for their comprehensive response to my questions. I also appreciate the difficult task of making additional measurements for a paper under review.

The authors have adequately addressed my concerns. I can recommend publication in Nature, pending the editor decision and responses from the other referees.

Referee #3 (Remarks to the Author):

This referee would like to sincerely thank the authors for their extensive and careful reply, for taking new data on a significantly larger device and most importantly, for carefully considering the difference between an intrinsic FSPT and an FSPT whose symmetry is equivalent to an equilibrium SPT. The new data on the Z_2 FPST and the EDSPT are extremely promising and the referee is hopeful that it might be possible to take slightly longer Floquet periods and to elevate these two results to the main text (see comments below). If the authors are able to carefully address my comments below, I would be happy to recommend the manuscript for publication in Nature.

1) At the moment, the intrinsically non-equilibrium FSPT (i.e. with only a Z_2 symmetry) and the quasi-periodically driven EDSPT feel a bit like an afterthought. I am wondering if the sentence "In addition, to demonstrate the general applicability of our digital simulation approach, we experimentally realize two further examples of nonequilibrium SPT phases with Z_2 and no microscopic symmetries, respectively (see Methods)," can somehow be moved earlier and more broadly, if the authors can perhaps present all three simulations i.e. $Z_2 \times Z_2$, Z_2 and EDSPT on the same footing? I think this would make the manuscript significantly stronger. I realize that this would perhaps require a bit of reworking the paper, but this referee thinks it would be worth it! Certainly, having the data on the Z_2 and EDSPT in the main text is important and demonstrates the strength and versatility of the simulation platform.

2) For the Z_2 FSPT and the EDSPT (extended data Fig 5,6), this referee is confused why the number of Floquet periods is significantly smaller than the $Z_2 \times Z_2$ SPT shown in Figure 2? In particular, for the quasi-periodic EDSPT, I would have expected that the key signatures occur in the edge correlation functions at the Fibonacci times (which is most naturally seen using logarithmically

spaced times); this makes it extremely difficult to see if the key signature of the EDSPT is actually being robustly realized in this system. This referee would very much be interested in seeing more data being taken in the EDSPT regime (and also the Z_2 FSPT regime).

Minor comments:

3) The phrase “spins in NV centers” in the abstract sounds a bit awkward, and I think at least one of the experiments focused on nuclear spins. Perhaps the authors could change this to simply “solid-state spin systems” and group ref 16,17 together as well.

4) Overall, I think the changes to the abstract are very nice, but it feels quite long and specific now. I would suggest dropping the whole sentence: “Unlike ...” and then just starting the next sentence with “We observe...”

5) I would recommend adding a citation to Phys. Rev. Lett. 119, 123601 (2017) together with ref 38 or 39-41, since it represents the first discussion of realizing an FSPT in a quantum simulator.

6) I would recommend dropping the phrase “...which we theoretically predict to exhibit an FSPT phase.”

7) Caption of Extended Data Fig. 5: I guess the authors are emphasizing the spin (Z_2) and discrete time translation symmetry (Z), but I am wondering it would be clearer to just mention the spin symmetry since that is the notation used in the rest of the manuscript. I’m also not precisely sure whether the symmetries should be a direct product or a semi-direct product?

Author Rebuttals to First Revision:

Response to Referee #3:

We sincerely thank the Referee for their careful reading of the revised manuscript and greatly appreciate their valuable suggestions, which have led us to carry out additional experiments on the Z_2 FSPT and ED- SPT models. Following their suggestions, we have clarified several important points and further improved the presentation. The detailed response to the Referee's comments is provided below.

Comment 1 of Referee #3: "This referee would like to sincerely thank the authors for their extensive and careful reply, for taking new data on a significantly larger device and most importantly, for carefully considering the difference between an intrinsic FSPT and an FSPT whose symmetry is equivalent to an equilibrium SPT. The new data on the Z_2 FPST and the EDSPT are extremely promising and the referee is hopeful that it might be possible to take slightly longer Floquet periods and to elevate these two results to the main text (see comments below). If the authors are able to carefully address my comments below, I would be happy to recommend the manuscript for publication in Nature."

Authors' response: We thank the Referee for judging that "The new data on the Z_2 FSPT and the EDSPT are extremely promising". We followed the Referee's suggestion and carried out additional experiments to extend the simulation times for both the Z_2 FSPT and the EDSPT models (see the following response for details and for our response to the suggestion to elevate these two results to the main text).

Comment 2 of Referee #3: "(1) At the moment, the intrinsically non-equilibrium FSPT (i.e. with only a Z_2 symmetry) and the quasi-periodically driven EDSPT feel a bit like an afterthought. I am wondering if the sentence "In addition, to demonstrate the general applicability of our digital simulation approach, we experimentally realize two further examples of nonequilibrium SPT phases with Z_2 and no microscopic symmetries, respectively (see Methods)," can somehow be moved earlier and more broadly, if the authors can perhaps present all three simulations, i.e. $Z_2 \times Z_2$, Z_2 and EDSPT on the same footing? I think this would make the manuscript significantly stronger. I realize that this would perhaps require a bit of reworking the paper, but this referee thinks it would be worth it! Certainly, having the data on the Z_2 and EDSPT in the main text is important and demonstrates the strength and versatility of the simulation platform."

Authors' response: We thank the Referee for this helpful suggestion. To put all three simulations on a more equal footing, in the revised manuscript we removed in the introduction the sentence “In addition, to demonstrate...” and explicitly mentioned “We successfully implement the dynamics of prototypical time-(quasi)periodic Hamiltonians with $Z_2 \times Z_2$, Z_2 , or no microscopic symmetries, and observe subharmonic temporal responses for the edge spins.” at the beginning of the third paragraph of the introduction section. In addition, we added a new section “Other nonequilibrium SPT phases” to the main text to promote to the discussion on the Z_2 FPST and EDSPT models. Due to space limitation, we did not move the Extended Figures 5 and 6 to the main text.

Comment 3 of Referee #3: “2) For the Z_2 FSPT and the EDSPT (extended data Fig 5,6), this referee is confused why the number of Floquet periods is significantly smaller than the $Z_2 \times Z_2$ SPT shown in Figure 2? In particular, for the quasi-periodic EDSPT, I would have expected that the key signatures occur in the edge correlation functions at the Fibonacci times (which is most naturally seen using logarithmically spaced times); this makes it extremely difficult to see if the key signature of the EDSPT is actually being robustly realized in this system. This referee would very much be interested in seeing more data being taken in the EDSPT regime (and also the Z_2 FSPT regime).”

Authors' response: We thank the Referee for raising this important question, which helped us clarify some confusing points and led us to carry out additional experiments to extend the simulation times for both the Z_2 FSPT and EDSPT models. First, we clarify that: 1) For the Z_2 FSPT model, the circuit depth for simulating one period is ten, which is larger than that for simulating the $Z_2 \times Z_2$ model (the circuit depth for one period in this case is six). Thus, for a given limited coherence time, the number of Floquet periods that we can simulate for the Z_2 FSPT model is in general smaller than that for the $Z_2 \times Z_2$ model; 2) For the EDSPT model, in the Extended Data Fig. 6 (c, d), what we really show on the x-axis is indeed the Fibonacci times, which are already on a logarithmic scale. In the revised manuscript, to avoid possible confusion, we have replaced the x-axis labels “t/T” with “Fibonacci times” for the Extended Data Fig. 6 (c, d).

Based on the Referee's suggestion, we have carried additional experiments to extend the number of Floquet periods for the Z_2 FSPT model and Fibonacci times for the EDSPT model. Now, for the Z_2 FSPT model, we obtained persistent subharmonic oscillations up to 22 cycles (compared to 8 cycles in the previous version of the manuscript). For the EDSPT model, we pushed our experiment to Fibonacci time 7 (compared to Fibonacci time 6 in the previous version of the manuscript), which corresponds to a circuit depth of 190. Due to the exponential scaling of circuit depth with Fibonacci time, the Fibonacci time 8 corresponds a circuit depth exceeding 300, which is beyond the capacity of our current device.

Comment 4 of Referee #3: “3) The phrase “spins in NV centers” in the abstract sounds a bit

awkward, and I think at least one of the experiments focused on nuclear spins. Perhaps the authors could change this to simply “solid-state spin systems” and group ref 16,17 together as well.”

Authors’ response: We thank the Referee for this constructive suggestion. Following the Referee’s suggestion, we have replaced “spins in NV centers” with “solid-state spin systems”, and grouped Refs. [16,17] together with Refs. [11-13].

Comment 5 of Referee #3: “4) Overall, I think the changes to the abstract are very nice, but it feels quite long and specific now. I would suggest dropping the whole sentence: “Unlike . . .” and then just starting the next sentence with “We observe. . .””

Authors’ response: We thank the Referee for this helpful suggestion. We removed this sentence as suggested.

Comment 6 of Referee #3: “5) I would recommend adding a citation to Phys. Rev. Lett. 119, 123601 (2017) together with ref 38 or 39-41, since it represents the first discussion of realizing an FSPT in a quantum simulator.”

Authors’ response: We thank the Referee for bringing this reference to our attention. We followed the Referee’s suggestion and added this reference (see Ref. [42]) together with Refs. [39-41].

Comment 7 of Referee #3: “6) I would recommend dropping the phrase “. . . which we theoretically predict to exhibit an FSPT phase.””

Authors’ response: We thank the Referee for this helpful suggestion. We have removed the phrase “. . . which we theoretically predict to exhibit an FSPT phase.”

Comment 8 of Referee #3: “7) Caption of Extended Data Fig. 5: I guess the authors are emphasizing the spin (Z_2) and discrete time translation symmetry (Z), but I am wondering it would be clearer to just mention the spin symmetry since that is the notation used in the rest of the manuscript. I’m also not precisely sure whether the symmetries should be a direct product or a semi-direct product?”

Authors’ response: We thank the Referee for this insightful suggestion. We agree with the Referee that it is better to just mention the spin symmetry and have revised the manuscript accordingly. As for the question of whether the symmetries should be a direct product or a semi-direct product, we think that “direct product” would be more accurate, since here the Z_2 spin symmetry is independent of the discrete time translation symmetry (Z).

In summary, we greatly appreciate the Referee's careful reading of the manuscript and their valuable suggestions. Following these suggestions, we have carried out additional experiments to extend the number of Floquet periods for both the Z_2 FPST and the EDSPT models. We made revisions to the manuscript accordingly and improved the presentation. We hope that this improved manuscript will satisfy the Referee and our work can be published in Nature.